# Compositional Entailment Learning for Hyperbolic Vision-Language Models

**Avik Pal**[1] **Max van Spengler**[1] **Guido Maria D'Amely di Melendugno**[2]
**Alessandro Flaborea**[234] **Fabio Galasso**[2] **Pascal Mettes**[1]

[1]University of Amsterdam    [2]Sapienza University of Rome    [3]ItalAI    [4]Procederai

## Abstract

Image-text representation learning forms a cornerstone in vision-language models, where pairs of images and textual descriptions are contrastively aligned in a shared embedding space. Since visual and textual concepts are naturally hierarchical, recent work has shown that hyperbolic space can serve as a high-potential manifold to learn vision-language representation with strong downstream performance. In this work, for the first time we show how to fully leverage the innate hierarchical nature of hyperbolic embeddings by looking beyond individual image-text pairs. We propose Compositional Entailment Learning for hyperbolic vision-language models. The idea is that an image is not only described by a sentence but is itself a composition of multiple object boxes, each with their own textual description. Such information can be obtained freely by extracting nouns from sentences and using openly available localized grounding models. We show how to hierarchically organize images, image boxes, and their textual descriptions through contrastive and entailment-based objectives. Empirical evaluation on a hyperbolic vision-language model trained with millions of image-text pairs shows that the proposed compositional learning approach outperforms conventional Euclidean CLIP learning, as well as recent hyperbolic alternatives, with better zero-shot and retrieval generalization and clearly stronger hierarchical performance. Code available at https://github.com/PalAvik/hycoclip.

## 1 Introduction

Vision-language modeling has witnessed rapid progress in recent years with innovative approaches such as CLIP (Radford et al., 2021) and ALIGN (Jia et al., 2021) using extensive vision-language data to train encoders for understanding visual and textual content simultaneously. Such encoders align visual scenes with textual descriptions in a shared high-dimensional Euclidean space, facilitating semantic understanding (Radford et al., 2021). While effective, conventional vision-language models only take a holistic approach to image-text representation learning, neglecting the intrinsic hierarchy and composition of elements within images. Indeed, a visual scene is commonly composed of multiple objects interacting with one another to form a precise context. See for example Fig. 1b with description: "*Mineral water* with *fresh herbs* in a *glass carafe* on a *garden table*". Individually, these objects provide limited semantic meaning. Only through the interactions between these do we understand the specific context of both the scene and its parts, characterizing the single entities (cf. Fig. 1a). This object-scene hierarchy is analogous to a parent-child connection in a discrete tree where broader concepts are closer to the root while specific concepts reside deeper in the tree. These tree-like structures cannot be well represented in Euclidean space due its polynomial volume growth (Matoušek, 1999), whereas hyperbolic geometry does accommodate the exponential growth of trees (Gromov, 1987), making it more suitable for representing hierarchies.

Recently, Desai et al. (2023) introduced MERU, a hyperbolic contrastive vision-language model. MERU projects Euclidean embeddings from image and text encoders onto hyperbolic space and enforces *inter-modal* (text to image) partial ordering (Vendrov et al., 2016) using an entailment loss (Ganea et al., 2018a; Le et al., 2019) when optimizing encoder weights. Such hyperbolic image-text alignment has demonstrated strong quantitative performance on zero-shot downstream tasks, as well as increased interpretability of the shared embedding space. They, however, ignore the

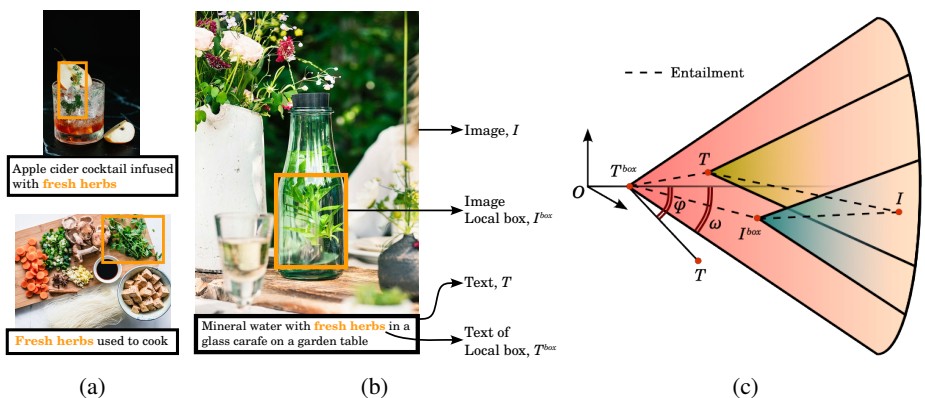

(a)  (b)  (c)

Figure 1: **Compositional Entailment Learning** for hyperbolic vision-language models. (a) same object appearing in different vision-language contexts (b) Visual-semantic ordering: $I$ (whole image) and $T$ (full caption) provide context to the more general $I^{box}$ (image local box) and $T^{box}$ (text local box). (c) This specific-general ordering between $(I, T), (I^{box}, T^{box}), (I, I^{box}), (T, T^{box})$ is enforced in hyperbolic space using entailment cones. The external angle $\phi$ of a specific concept ($T$) is pushed to be within the aperture threshold $\eta\omega$ of the general concept ($T^{box}$).

*intra-modal* hierarchical compositions of image-text pairs. Indeed, there is hierarchical semantics in language (Everaert et al., 2015), which has been leveraged to embed textual data in hyperbolic space (Dhingra et al., 2018). In the vision domain, work by Ge et al. (2023) uses object-centric scene hierarchies to learn a hyperbolic space where visually similar objects are clustered near the origin and scenes consisting of them are descendants. Zhong et al. (2022) propose RegionCLIP that only learns regional representations using contrastive learning and knowledge distillation. These prior works beg the question of what strategy can be adapted to compound the individual benefits of the inter-modal hierarchy and the two intra-modal hierarchies to encompass *scene and region* level understanding.

To this end, we introduce Hyperbolic Compositional CLIP (HyCoCLIP), a contrastive learning method that accounts for compositional orders in both inter-modal and intra-modal settings in hyperbolic space. We approach the problem by using explicit hierarchies while training the encoders. This hierarchy is constituted of object crops (*image boxes*) within an image and corresponding nouns/phrases (*text boxes*) within the text as broader concepts of the whole image-text concept. We outline a robust hierarchical learning objective by using both entire images and image boxes, as well as complete captions and text boxes. This strategy involves both inter-modal hierarchies, where text generally provides broader context than images, and intra-modal hierarchies, where we consider the "boxes" more general than the complete image. In the hierarchical spatial representation, broader concepts are embedded near the origin of the metric space, while more fine-grained concepts are positioned towards the border, akin to tree graphs, see Fig. 1c.

We show that HyCoCLIP outperforms CLIP and MERU on zero-shot image classification and is competitive on zero-shot retrieval and object detection when trained on a 20M pre-training dataset. Additionally, we show that HyCoCLIP improves on hierarchical classification tasks compared to the baselines and that its representation space is more interpretable and hierarchically aligned. Our contributions are summarized as follows: (1) We introduce HyCoCLIP for learning vision-language representations in a shared hyperbolic space using scene compositions that are semantically and hierarchically aligned. (2) We propose Compositional Entailment Learning, where image-text compositions are optimized through hyperbolic contrastive and entailment cone losses. (3) We demonstrate empirically that HyCoCLIP is more hierarchically aware and is highly competitive to existing vision-language models.

## 2 HYPERBOLIC COMPOSITIONAL CLIP - HYCOCLIP

We propose a compositional learning scheme that enforces the semantic alignment of latent representations in the hyperbolic space, explicitly modeling intra- and inter-modal relationships of visual and language data by leveraging their joint hierarchical nature. Here, we first provide a short background

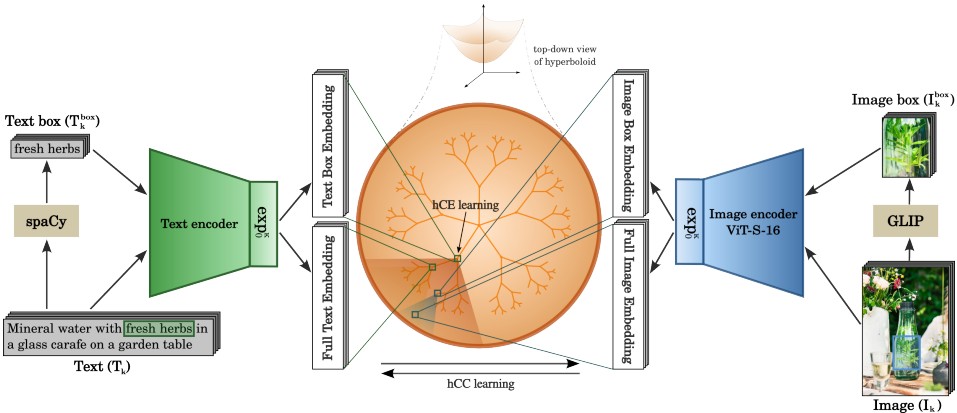

Figure 2: **An overview of HyCoCLIP.** Text and image boxes are extracted offline from image-text datasets (sides). Next, HyCoCLIP's encoder modules embed the images and texts, projecting the representations in the hyperbolic latent space. HyCoCLIP preserves the inter-modal and intra-modal relationships by accommodating broader/finer concepts close to the center/border and by using entailment cones to give an interpretable structure to the learned latent space (cf. Fig. 1c).

with the required hyperbolic functions to make such compositional learning possible. Afterward, we outline our compositional encoding of image-text pairs.

## 2.1 BACKGROUND

Hyperbolic geometry is a non-Euclidean geometry characterized by a constant negative curvature. The resulting space has the desirable property that volumes of subsets can grow exponentially as a function of their radius, making it an ideal choice for learning representations of data with an inherent hierarchical or tree-like structure (Sarkar, 2011; Nickel & Kiela, 2017; Krioukov et al., 2010). While several isometric models are used in literature for modeling hyperbolic space, we limit our background discussion to the Lorentz (or hyperboloid) model used in this work and refer to Cannon et al. (1997); Peng et al. (2022) for detailed information on the other models.

The Lorentz model, denoted by $\mathbb{L}^n$, is an $n$-dimensional manifold represented as the upper sheet of a two-sheeted hyperboloid in $(n+1)$-dimensional Minkowski spacetime. For each vector $\mathbf{p} \in \mathbb{R}^{n+1}$, the first dimension is taken as the *time*-axis, denoted $p_0$, and the remaining $n$ dimensions as the *spatial*-coordinates, denoted $\tilde{\boldsymbol{p}} \in \mathbb{R}^n$. This model is described as

$$\mathbb{L}^n = \left\{ \mathbf{p} \in \mathbb{R}^{n+1} : \langle \mathbf{p}, \mathbf{p} \rangle_\mathbb{L} = -\frac{1}{\kappa}, p_0 = \sqrt{1/\kappa + \|\tilde{\boldsymbol{p}}\|^2}, \kappa > 0 \right\}, \tag{1}$$

where $-\kappa \in \mathbb{R}$ is the curvature of the space and $\langle ., . \rangle_\mathbb{L}$ is the Lorentzian inner product defined for $\mathbf{p}, \mathbf{q} \in \mathbb{L}^n$ as

$$\langle \mathbf{p}, \mathbf{q} \rangle_\mathbb{L} = -p_0 q_0 + \langle \tilde{\boldsymbol{p}}, \tilde{\boldsymbol{q}} \rangle_\mathbb{E}, \tag{2}$$

with $\langle ., . \rangle_\mathbb{E}$ denoting the Euclidean inner product. The Lorentzian distance between two points in $\mathbb{L}^n$ is the length of the shortest path (*geodesic*) connecting them, which can be computed as

$$d_\mathbb{L}(\mathbf{p}, \mathbf{q}) = \sqrt{1/\kappa} \cdot \cosh^{-1}\left( -\kappa \langle \mathbf{p}, \mathbf{q} \rangle_\mathbb{L} \right), \quad \mathbf{p}, \mathbf{q} \in \mathbb{L}^n. \tag{3}$$

This metric induces the Lorentzian norm $\|\mathbf{p}\|_\mathbb{L} = \langle \mathbf{p}, \mathbf{p} \rangle_\mathbb{L}$. The tangent space $T_\mathbf{p}\mathbb{L}^n$ is well-defined for all the points $\mathbf{p} \in \mathbb{L}^n$, and the exponential map represents the projecting map from the tangent space to the hyperboloid. Given a point $\mathbf{v} \in T_\mathbf{p}\mathbb{L}^n$ the exponential map can be computed as

$$\exp_\mathbf{p}^\kappa(\mathbf{v}) = \cosh(\sqrt{\kappa}\|\mathbf{v}\|_\mathbb{L})\mathbf{p} + \frac{\sinh(\sqrt{\kappa}\|\mathbf{v}\|_\mathbb{L})}{\sqrt{\kappa}\|\mathbf{v}\|_\mathbb{L}}\mathbf{v}. \tag{4}$$

Such a map can be used to move from Euclidean space to hyperbolic space by considering Euclidean vectors to be tangent vectors at the origin $\mathbf{0} = (\sqrt{1/\kappa}, 0, \ldots, 0)^T$ of the hyperbolic space and using $\exp_\mathbf{0}^\kappa$ to project these onto the hyperboloid (Khrulkov et al., 2020).

## 2.2 COMPOSITIONAL ENTAILMENT LEARNING

We strive to learn the hierarchical compositional relations of images, boxes, and textual descriptions. Our idea is based on the following observation: the content inside a box of an image is hierarchically more general than the entire image. While counter-intuitive at first glance, Fig. 1b shows why this is the case: the box shows an object and the entire image additionally shows the context in which the object occurs, making it a semantically more specific scenario. From a hyperbolic perspective, semantically general/broad concepts are embedded closer to the origin, while more fine-grained concepts are positioned towards the border, akin to tree graphs (cf. Fig. 1c).

In this work, we are given a dataset $D = \{(I_k, T_k)\}_{k=1}^K$ of $K$ image-text pairs. Our goal is to train image and text encoders with a shared embedding space to align the visual and semantic inputs. The method is summarized in Fig. 2. Let $(I_k^{\text{box}}, T_k^{\text{box}})$ be the local box with a short description from an image-text pair obtained following the automated procedure detailed in Appendix A. We propose a Compositional Entailment Learning objective in hyperbolic space to optimize the hierarchical compositions. Our approach consists of two parts, namely a compositional contrastive loss and a compositional entailment loss which we discuss next.

**Hierarchical Compositional Contrastive (hCC) learning.** Image-text models commonly rely on contrastive objectives to align and distribute the multi-modal data. In our approach, we rely on hyperbolic embeddings to align visuals and text. Let $f_I(\cdot)$ and $f_T(\cdot)$ denote arbitrary encoders for the image and text inputs respectively. And, let $g_I(I_k) = \exp_{\mathbf{0}}^\kappa(f_I(I_k))$ and $g_T(T_k) = \exp_{\mathbf{0}}^\kappa(f_T(T_k))$ denote the hyperbolic representation of image $I_k$ and textual description $T_k$ respectively. To compute the contrastive loss over image-text pairs in a batch $B$, we take the negative Lorentzian distance as our similarity metric and formulate it with the softmax, using temperature $\tau$, for a batch of size $(B)$ containing images $(I)$ and text $(T)$ as

$$L_{cont}^*(I, T) = -\sum_{i \in B} \log \frac{\exp\big(d_{\mathbb{L}}(g_I(I_i), g_T(T_i))/\tau\big)}{\sum_{k=1, k \neq i}^B \exp\big(d_{\mathbb{L}}(g_I(I_i), g_T(T_k))/\tau\big)}, \tag{5}$$

where negatives for an image are picked from the texts in the batch. Similarly, we can define the loss when picking negatives for a text from images in the batch as $L_{cont}^*(T, I)$. To extend such a contrastive setup with our image-text compositions, we have to consider that due to the generalized information in a box, different images in a batch can have similar box-level information. To avoid unwanted negatives in a batch, we only contrast the box image with other entire images, and vice versa which have specific information. This avoids negative alignment between image-box pairs and boxes from different images. The final hierarchical Compositional Contrastive (hCC) loss is formulated as

$$hCC(I, T, I^{box}, T^{box}) = \frac{1}{4}\Big(\underbrace{L_{cont}^*(I, T) + L_{cont}^*(T, I)}_{\text{specific-info contrast}} + \underbrace{L_{cont}^*(I^{box}, T) + L_{cont}^*(T^{box}, I)}_{\text{general-info contrast}}\Big).$$

$$\tag{6}$$

**Hierarchical Compositional Entailment (hCE) learning.** Ganea et al. (2018a) introduced hyperbolic entailment cones that generalize the idea of partial order embeddings (Vendrov et al., 2016) by using the inherent hierarchical structure of the hyperbolic space. Entailment cones define a region $\Re_q$ for every possible point $\mathbf{q}$ in the space such that all points $\mathbf{p} \in \Re_q$ are semantically linked to $\mathbf{q}$ as its child concepts. As such, points in $\Re_q$ are expected to contain specific information for the general concept $\mathbf{q}$. Considering the Lorentz model $\mathbb{L}^n$, the half-aperture of these conical regions ($\Re_q$) is formulated by Le et al. (2019); Desai et al. (2023) as

$$\omega(\mathbf{q}) = \sin^{-1}\left(\frac{2K}{\sqrt{\kappa}\|\tilde{\boldsymbol{q}}\|}\right), \tag{7}$$

where $-\kappa$ is the curvature of the space and a constant $K = 0.1$ is set to limit values near the origin (see Ganea et al. (2018a)). The aperture inversely depends on the norm $\|\tilde{\boldsymbol{q}}\|$. Inferring from this, a general concept with a wider aperture would lie closer to the origin. A specific concept would have a narrower aperture and lie further from the origin in the hyperbolic space.

To learn partial orders in this space, specific concepts $\mathbf{p}$ must be pushed to be within the aperture $\omega(\mathbf{q})$. This is done by penalizing encoders with the angular residual of outward point $\mathbf{p}$ having an exterior angle $\phi(\mathbf{p}, \mathbf{q})$ as shown in Fig. 1c. This is formulated by Le et al. (2019); Desai et al. (2023) as

$$L_{ent}(\mathbf{p}, \mathbf{q}) = \max(0, \phi(\mathbf{p}, \mathbf{q}) - \omega(\mathbf{q})), \quad (8)$$

where the exterior angle is given by,

$$\phi(\mathbf{p}, \mathbf{q}) = \cos^{-1}\left(\frac{p_0 + q_0\kappa\langle\mathbf{p}, \mathbf{q}\rangle_{\mathbb{L}}}{\|\tilde{\mathbf{q}}\|\sqrt{(\kappa\langle\mathbf{p}, \mathbf{q}\rangle_{\mathbb{L}})^2 - 1}}\right). \quad (9)$$

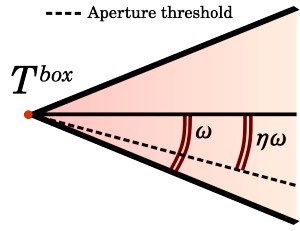

Figure 3: **Aperture threshold** $\eta$ scaling the aperture $\omega$ to increase or decrease the width of the entailment cone.

However, intuitively observing the entailment loss presented in Eq. 8 shows that this loss would push any outward point $\mathbf{p}$ only towards the $\Re_q$ region's border. Hence, we add a threshold to the half-aperture $\omega(\mathbf{q})$ effectively making it flexible to accommodate $\mathbf{p}$ at various spatial distances from $\mathbf{q}$, see Fig. 3. We reformulate Eq. 8 with half-aperture threshold $\eta$ as

$$L^*_{ent}(\mathbf{p}, \mathbf{q}) = \max(0, \phi(\mathbf{p}, \mathbf{q}) - \eta\omega(\mathbf{q})). \quad (10)$$

Entailment cones enable us to enforce the hierarchical image-text relations given by the compositions. We formulate the Hierarchical Compositional Entailment (hCE) loss by considering that images and textual descriptions are not identical, but that text precedes image, akin to Desai et al. (2023). We additionally consider the relation *whole* $\Rightarrow$ *box* for both images and texts. Hence, the hCE loss would comprise both image-text inter-modality entailments and text-text, image-image intra-modality entailments as

$$hCE(I, T, I^{box}, T^{box}) = \underbrace{L^*_{ent}(I^{box}, T^{box}) + L^*_{ent}(I, T)}_{\text{inter-modality entailment}} + \underbrace{L^*_{ent}(I, I^{box}) + L^*_{ent}(T, T^{box})}_{\text{intra-modality entailment}}.$$

$$(11)$$

In Fig. 1c we visualize how the image-text compositions should be organized in hyperbolic compositional entailment.

**Hierarchical Compositional (hC) learning.** We aggregate the losses to form the overall hierarchical Compositional (hC) loss for HyCoCLIP by taking a weighted sum of the two loss components:

$$hC = hCC + \gamma hCE. \quad (12)$$

In Appendix B, we detail all hyperparameters, thresholds, and further implementation details.

**Computational complexity.** Our approach enables us to double the amount of visual and textual data to learn from. The training time scales linearly with the increase in training volume; for ViT-B/16, HyCoCLIP requires 73 hours of training, compared to 46 hours for MERU and 45 hours for CLIP. We note that our method inference maintains the same efficiency as CLIP and MERU and allows for scalable deployment.

## 3 EXPERIMENTS

### 3.1 BENCHMARK

**Datasets** We develop our models using grounded vision-language pairs. This could be human-annotated such as the Localized narratives subset of Open Images (Pont-Tuset et al., 2020) or the Flickr30K Entities dataset (Plummer et al., 2015). However, the sizes of these datasets are fairly limited considering the intensive efforts of manual labelling. Hence, we depend on automatic grounded information generated by pre-trained phrase grounding models. Several large-scale grounded language-vision datasets are publicly available by Li et al. (2023) and Peng et al. (2023). We train our models using the large-scale training corpus - Grounded Image-Text Pairs (GRIT) dataset (Peng et al., 2023) containing 20.5 million grounded vision-language pairs which are processed from the even larger COYO-700M (Byeon et al., 2022) dataset. Information on the grounding procedure is added in Appendix A. We similarly use the grounding procedure on the RedCaps dataset (Desai et al., 2021) originally used to train MERU. Additionally, we use the smaller-scale grounded Conceptual Captions 3M (CC3M) (Li et al., 2023; Sharma et al., 2018) dataset for hyperparameter search.

Table 1: **Zero-shot image classification evaluation.** † denotes reproduced results from MERU. When using boxes during pre-training, numbers in squared brackets represent the additional box-pairs counts. For RedCaps, we find results for CLIP and MERU consistent with Desai et al. (2023) even when trained with a smaller batch size. **Bold-face** numbers are the best overall performances, while underlined figures are the best within all the models sharing the same ViT backbone. Our method outperforms baselines on 15 out of the 16 evaluation datasets.

| | | w/ boxes | samples (M) | ImageNet | CIFAR-10 | CIFAR-100 | SUN397 | Caltech-101 | STL-10 | Food-101 | CUB | Cars | Aircraft | Pets | Flowers | DTD | EuroSAT | RESISC45 | Country211 |
|---|---|---|---|---|---|---|---|---|---|---|---|---|---|---|---|---|---|---|---|
| | | | | | | General datasets | | | | | | Fine-grained datasets | | | | | | Misc. datasets | | |
| **RedCaps** | | | | | | | | | | | | | | | | | | | | |
| ViT S/16 | CLIP† | ✗ | 11.4 | 32.5 | 66.7 | 35.8 | 26.7 | 60.8 | 89.8 | 72.5 | 29.8 | 11.1 | 1.3 | 72.5 | 44.9 | 16.4 | 30.1 | 27.7 | 5.0 |
| | CLIP | ✓ | 11.4 [6.3] | 30.2 | 76.5 | 42.4 | 25.8 | 62.3 | 89.5 | 69.6 | 25.7 | 8.5 | 2.2 | 65.3 | 38.6 | 13.6 | 36.6 | 28.5 | 4.6 |
| | MERU† | ✗ | 11.4 | 31.4 | 65.9 | 35.2 | 26.8 | 58.1 | 89.3 | 71.4 | 29.0 | 8.3 | 1.6 | 71.0 | 40.9 | 17.0 | 29.9 | 29.3 | 4.7 |
| | MERU | ✓ | 11.4 [6.3] | 29.9 | 76.4 | 39.9 | 26.6 | 62.3 | 89.5 | 68.4 | 25.4 | 8.9 | 1.2 | 67.2 | 37.6 | 13.0 | 30.5 | 27.6 | 4.3 |
| | HyCoCLIP | ✓ | 5.8 [6.3] | 31.9 | 77.4 | 37.7 | 27.6 | 64.5 | 90.9 | 71.1 | 28.8 | 9.7 | 1.1 | 70.5 | 41.4 | 13.4 | 22.7 | 30.7 | 4.4 |
| **GRIT** | | | | | | | | | | | | | | | | | | | | |
| ViT S/16 | CLIP | ✗ | 20.5 | 36.7 | 70.2 | 42.6 | 49.5 | 73.6 | 89.7 | 44.7 | 9.8 | 6.9 | 2.0 | 44.6 | 14.8 | 22.3 | 40.7 | 40.1 | 5.1 |
| | CLIP | ✓ | 20.5 [35.9] | 36.2 | 84.2 | 54.8 | 46.1 | 74.1 | 91.6 | 43.2 | 11.9 | 6.0 | 2.5 | 45.9 | 18.1 | 24.0 | 32.4 | 35.5 | 4.7 |
| | MERU | ✗ | 20.5 | 35.4 | 71.2 | 42.0 | 48.6 | 73.0 | 89.8 | 48.8 | 10.9 | 6.5 | 2.3 | 42.7 | 17.3 | 18.6 | 39.1 | 38.9 | 5.3 |
| | MERU | ✓ | 20.5 [35.9] | 35.0 | 85.0 | 54.0 | 44.6 | 73.9 | 91.6 | 41.1 | 10.1 | 5.6 | 2.2 | 43.9 | 15.9 | 24.5 | 39.3 | 33.5 | 4.8 |
| | HyCoCLIP | ✓ | 20.5 [35.9] | 41.7 | 85.0 | 53.6 | 52.5 | 75.7 | 92.5 | 50.2 | 14.7 | 8.1 | 4.2 | 52.0 | 20.5 | 22.3 | 33.8 | 45.7 | 5.2 |
| ViT B/16 | CLIP | ✗ | 20.5 | 40.6 | 78.9 | 48.3 | 53.0 | 76.7 | 92.4 | 48.6 | 10.0 | 9.0 | 3.4 | 45.9 | 21.3 | 23.4 | 37.1 | 42.7 | 5.7 |
| | MERU | ✗ | 20.5 | 40.1 | 78.6 | 49.3 | 53.0 | 72.8 | 93.2 | 51.5 | 11.9 | 8.6 | 3.7 | 48.5 | 21.2 | 22.2 | 31.7 | 44.2 | 5.6 |
| | HyCoCLIP | ✓ | 20.5 [35.9] | 45.8 | 88.8 | 60.1 | 57.2 | 81.3 | 95.0 | 59.2 | 16.4 | 11.6 | 3.7 | 56.8 | 23.9 | 29.4 | 35.8 | 45.6 | 6.5 |

**Baseline Comparisons** We compare HyCoCLIP against CLIP and MERU. We reproduce the CLIP and MERU models by training on the RedCaps dataset, reducing the batch size to 768 to fit on our available compute. We further retrain CLIP and MERU from scratch on the GRIT dataset. To fairly evaluate the impact of including image-text boxes, we also retrain CLIP and MERU when image-text boxes are included as additional pre-training samples.

## 3.2 Downstream Tasks

To assess performance, we evaluate HyCoCLIP on several downstream tasks. For zero-shot image classification, the label set is fitted to multiple prompts which are embedded using the text encoder and then averaged to obtain a single embedding per label. The closest text embedding is picked from the collection as the predicted class for an image. We report the model's accuracy on 16 image classification datasets. Similarly, we assess our method on zero-shot retrieval tasks to determine if complex concepts, like scenes and captions, are accurately preserved in the representation space. Further, we evaluate the models on object detection task to analyze the regional understanding of HyCoCLIP. We also evaluate the hierarchical nature of HyCo-CLIP using multiple hierarchical metrics. Additionally, we assess the scene understanding capability of HyCoCLIP on two compositional benchmarks - VL-Checklist (Zhao et al., 2022) and VG Attribution (Yüksekgönül et al., 2023).

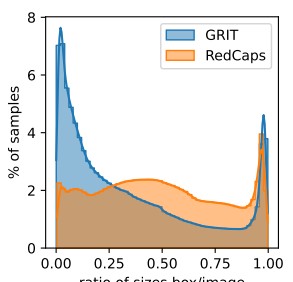

Figure 4: **Histogram of ratios** of box area *wrt* the full image for GRIT and Red-Caps. The latter reports generally larger crops, indicating lower precision in grounding concepts.

**Zero-shot image classification** From Table 1, we find our reproduced results for CLIP and MERU are fairly consistent with Desai et al. (2023) even when trained with smaller batch size on RedCaps. On grounding RedCaps and filtering noise, we notice only 5.8 million image-text pairs are retained containing 6.3 million boxes. Training the baselines with these additional boxes also demonstrates reduced performance. Alternatively, the quantity of data is significantly higher for GRIT with 20.5 million image-text pairs and 35.9 million boxes in total. To differentiate between the datasets, we compare the ratio of the box area with the image area of all data points and plot a histogram in Fig. 4. A lower ratio signifies that phrases have more localized information in the image and constitute a better semantic parent for the whole image. This is evident for GRIT while boxes generated for RedCaps do not seem to localize well. We, therefore, recommend GRIT over RedCaps for grounded pre-training and thus report the results of models pre-trained on GRIT. We find that HyCoCLIP

Table 2: **Zero-shot retrieval, detection, and hierarchical classification.** HyCoCLIP performs best in image retrieval, and hierarchical classification and is competitive in text retrieval. **Bold** figures indicate the best results for each ViT backbone.

| Vision encoder | Model | w/ boxes | Text retrieval | | | | Image retrieval | | | | Hierarchical metrics | | | | |
|---|---|---|---|---|---|---|---|---|---|---|---|---|---|---|---|
| | | | COCO | | Flickr | | COCO | | Flickr | | WordNet | | | | |
| | | | R@5 | R@10 | R@5 | R@10 | R@5 | R@10 | R@5 | R@10 | TIE($\downarrow$) | LCA($\downarrow$) | $J(\uparrow)$ | $P_H(\uparrow)$ | $R_H(\uparrow)$ |
| ViT S/16 | CLIP | ✗ | 69.3 | 79.1 | **90.2** | **95.2** | 53.7 | 65.2 | 81.1 | 87.9 | 4.02 | 2.39 | 0.76 | 0.83 | 0.84 |
| | CLIP | ✓ | 60.7 | 71.8 | 84.2 | 91.3 | 47.1 | 58.6 | 73.1 | 82.1 | 4.03 | 2.38 | 0.76 | 0.83 | 0.83 |
| | MERU | ✗ | 68.8 | 78.8 | 89.4 | 94.8 | 53.6 | 65.3 | 80.4 | 87.5 | 4.08 | 2.39 | 0.76 | 0.83 | 0.83 |
| | MERU | ✓ | **72.7** | **81.9** | 83.5 | 90.1 | 46.6 | 58.3 | 60.0 | 71.7 | 4.08 | 2.39 | 0.75 | 0.83 | 0.83 |
| | HyCoCLIP | ✓ | 69.5 | 79.5 | 89.1 | 93.9 | **55.2** | **66.6** | **81.5** | **88.1** | **3.55** | **2.17** | **0.79** | **0.86** | **0.85** |
| ViT B/16 | CLIP | ✗ | 71.4 | 81.5 | **93.6** | **96.9** | 57.4 | 68.5 | 83.5 | 89.9 | 3.60 | 2.21 | 0.79 | 0.85 | 0.85 |
| | MERU | ✗ | **72.3** | 82.0 | 93.5 | 96.2 | 57.4 | 68.6 | 84.0 | 90.0 | 3.63 | 2.22 | 0.78 | 0.85 | 0.85 |
| | HyCoCLIP | ✓ | 72.0 | **82.0** | 92.6 | 95.4 | **58.4** | **69.3** | **84.9** | **90.3** | **3.17** | **2.05** | **0.81** | **0.87** | **0.87** |

performs best across a wide range of datasets and settings when pre-training is done on GRIT. We especially note the performance on ImageNet, where we obtain an accuracy of 45.8% compared to 40.1% (MERU) and 40.6% (CLIP). Interestingly, adding image-text boxes to CLIP and MERU training does not improve performance, despite nearly doubling the training samples.

**Zero-shot retrieval** For the retrieval task, the top-k image/text embeddings are picked from a collection for input text/image embedding based on the distance score (Eq. 3). We perform this task zero-shot on the COCO validation set (Lin et al., 2014) and the Flickr30K test set (Young et al., 2014; Karpathy & Fei-Fei, 2015). We show the retrieval results in Table 2. We find that our method performs slightly worse on Flickr text retrieval while demonstrating increased performance on image retrieval over CLIP and MERU. We also note a significant decrease in the performance of CLIP and MERU when adding local information. These results further highlight the need for our approach. Naively adding these boxes as additional samples is not effective because the boxes are often without broader context, and the text is highly generic compared to the whole images. Only by optimizing for their hierarchical compositional nature as done in our approach is it possible to get better performance. Our method aims to obtain a hierarchically aligned representation space, but this is not necessarily beneficial for the task of retrieval, where proximity of text and image embeddings is key. Regardless, our approach remains highly competitive.

**Hierarchical Classification** A characteristic feature of hyperbolic spaces is their ability to represent hierarchical structures present in data. We evaluate our models for this property on several hierarchical classification metrics (Kosmopoulos et al., 2015) described in Appendix C. We use the WordNet hierarchy (Miller, 1994) of the ImageNet class labels (Deng et al., 2009; Russakovsky et al., 2015) for the hierarchical classification task. The image classification setup is kept similar and the final scores are averaged over the validation set. Table 2 reports the results of HyCoCLIP and other baselines on these metrics. We observe a consistent improvement, confirming that the hierarchy of the class labels is better represented in its embedding space.

**Zero-shot object detection** We utilize pre-trained vision-language models to recognize proposed object regions. Specifically, we evaluate the scenario where ground-truth bounding boxes from the COCO detection dataset are used as region proposals and predict the correct categories with a setup similar to image classification. We compare our method with RegionCLIP (Zhong et al., 2022) whose vision encoder (ResNet50x4) was trained on CC3M with a frozen text encoder (originally trained on CLIP400M). We report the average precision (AP) on the 17 novel categories data split (Bansal et al., 2018). As shown in Table 3, HyCoCLIP outperforms the baselines, surpassing RegionCLIP on the novel categories. We believe this highlights a key advantage of our approach—its ability to leverage inherent hierarchies for more effective semantic concept alignment.

Table 3: **Zero-shot object detection** with ground-truth boxes evaluated on COCO 17 novel categories split (Bansal et al., 2018). HyCoCLIP shows the best average precision (AP).

| Model | AP |
|---|---|
| CLIP | 51.2 |
| MERU | 55.8 |
| RegionCLIP | 65.2 |
| HyCoCLIP | **68.5** |

Table 4: **Ablation study on loss terms during pre-training** of HyCoCLIP-ViT-S/16 on grounded CC3M evaluated for image classification and Flickr image/text retrieval. Lower accuracy/R@5 indicates a more influential loss term.

| Pre-training losses | classification | | | retrieval | |
| --- | --- | --- | --- | --- | --- |
| | ImageNet | Food-101 | Mean(16) | Text | Image |
| HyCoCLIP | 16.7 | 10.6 | 22.3 | 56.2 | 46.4 |
| hCC loss | | | | | |
| $-L^*_{cont}(I, T)$ | 16.0 | 9.4 | 22.5 | 55.2 | 46.2 |
| $-L^*_{cont}(T, I)$ | 16.1 | 10.2 | 22.2 | 55.2 | 45.4 |
| $-L^*_{cont}(I^{box}, T)$ | 13.8 | 8.7 | 19.3 | 49.1 | 42.9 |
| $-L^*_{cont}(T^{box}, I)$ | 15.2 | 7.6 | 20.4 | 55.9 | 44.5 |
| hCE loss | | | | | |
| $-L^*_{ent}(I, T)$ | 14.9 | 9.9 | 21.8 | 54.3 | 46.1 |
| $-L^*_{ent}(I^{box}, T^{box})$ | 16.1 | 9.3 | 22.3 | 54.8 | 45.4 |
| $-L^*_{ent}(I, I^{box})$ | 16.1 | 10.1 | 21.5 | 55.0 | 45.6 |
| $-L^*_{ent}(T, T^{box})$ | 16.3 | 9.2 | 22.3 | 55.6 | 45.1 |

Table 5: **Scene understanding** evaluation. HyCoCLIP show better performance on both benchmarks indicating good object comprehension of the visual scene.

| Model | VL-CO | VG-A |
| --- | --- | --- |
| CLIP | 49.3 | 63.3 |
| MERU | 50.5 | 61.8 |
| HyCoCLIP | **59.8** | **68.4** |

Table 6: **Ablation study on batch size** shows saturation after 768.

| Batch size | ImageNet |
| --- | --- |
| 512 | 11.1 |
| 640 | 11.3 |
| 768 | 12.2 |
| 896 | 12.2 |
| 1024 | 12.1 |
| 1536 | 12.1 |

**Scene Understanding** Hyperbolic embeddings have previously demonstrated enhanced spatial awareness (Ibrahimi et al., 2024). We expect HyCoCLIP to be provisioned with further localized object/noun information in both vision and language and improve upon such aspects. The setup for these benchmarks is the same, for a given image the model has to pick between the correct caption and a hard negative caption. For more information on the methods used to generate the hard negative captions, we refer to Appendix E. From Table 5, we see that CLIP and MERU give near-random performance for the VL-Checklist-Object (VL-CO) (Zhao et al., 2022) benchmark in which object information in the captions is perturbed. HyCoCLIP improves considerably on these experiments reaching 60% accuracy indicating good object comprehension of the visual scene. HyCoCLIP also performs well on VG-Attribution (VG-A) (Yüksekgönül et al., 2023) reporting a mean accuracy of 68.4% surpassing other methods. We refer to Appendix E for further analysis.

## 3.3 ABLATION STUDY

**Pre-training loss terms** We examine the impact of the terms in hCC (Eq. 6) and hCE (Eq. 11) losses by pre-training the model several times, each time turning off a single loss term. We use the grounded CC3M dataset and train for 40k steps. Table 4 shows the results of this experiment. A lower accuracy and recall on image classification and retrieval respectively, indicate a higher influence of corresponding loss term. For hCC loss, we find that our hypothesis of contrasting the generalized

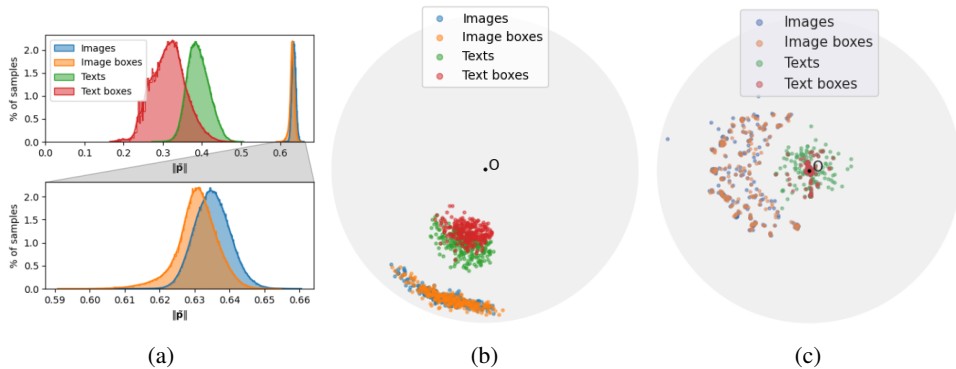

Figure 5: **Visualizing the learned hyperbolic space of HyCoCLIP in lower dimensions** using samples from GRIT. (a) distribution of embedding distances from the origin, HyCoCLIP embeds text data closer to the origin *wrt* the images and boxes samples with a smaller radius *wrt* full images/captions. On the right, (b) HoroPCA and (c) CO-SNE visualizations of the latent space in $\mathbb{L}^2$.

information in boxes against entire images and text is indeed beneficial. For hCE loss, we see that the terms entailing the image ($I$) are most influential.

**Scaling w.r.t batch size**   We train our models using a batch size 768 according to available compute (Appendix B). To study the influence of this hyperparameter, we train our primary baseline MERU-ViT-S using CC3M for various batch sizes and report their zero-shot performance on ImageNet classification. Table 6 indicates no empirical benefits when working with larger batch sizes. In the contrastive setting, the number of positives grows linearly, and the number of negatives grows quadratically in a batch. When using softmax, the ratio of positives to negatives affects loss functions differently depending on the type of similarity metric that is being used. This can explain the difference in batch size behavior of our approach. The saturation of softmax loss with increasing batch size has been previously discussed by Zhai et al. (2023), and the entailment loss may also contribute to this early saturation.

## 4   ANALYZING THE HYPERBOLIC SPACE

**Visualizing the learned hyperbolic space**   We visualize the learned hyperbolic space in lower dimensions to see if the image, text, and corresponding box embeddings are distributed in a proper semantic hierarchy. To this end, we plot the distribution of the spatial norms of 128k random samples of training data in a histogram. Furthermore, we use *HoroPCA* (Chami et al., 2021) for reducing the dimensionality for 200 image-text pairs along with their boxes. Lastly, we extract 50 principal components to suppress noise and use CO-SNE (Guo et al., 2022) to bring the embeddings to the low-dimensional space.

Fig. 5a shows that the embedding distributions of texts and their corresponding boxes are well separated, while images and their box representations display similar norms. This spatial contraction in image embeddings arises from the convergence of contrastive loss within a confined entailment cone, as noted by Ramasinghe et al. (2024). Furthermore, many image boxes are almost identical to the full image (cf. Fig. 4), making it challenging for the network to differentiate between them. Nonetheless, the bottom plot in Fig. 5a shows that the box embeddings distribute closer to the origin, thus displaying hierarchical ordering. From Fig. 5b and 5c, we observe the semantic separation in the two principal components of HoroPCA and in the 2D space formed with CO-SNE, indicating an apparent hierarchy between the components.

**Interpolating between points in hyperbolic space**   We interpolate the geodesic connecting an image (source) with another image (target) and also with the origin similar to Desai et al. (2023), which have been visualized on the bottom-right of Fig. 6. This intuitively represents traversing between nodes in a discrete tree. This is useful in visualizing the ancestors of any given image and qualitatively verifying the hierarchical properties of the learned hyperbolic space. We do the shortest path traversal in the tangent space details of which are in Appendix G. We use grounded Flickr30K (Li et al., 2023) to generate the collection of representations of images, texts, and corresponding boxes. Fig. 6 shows the result of

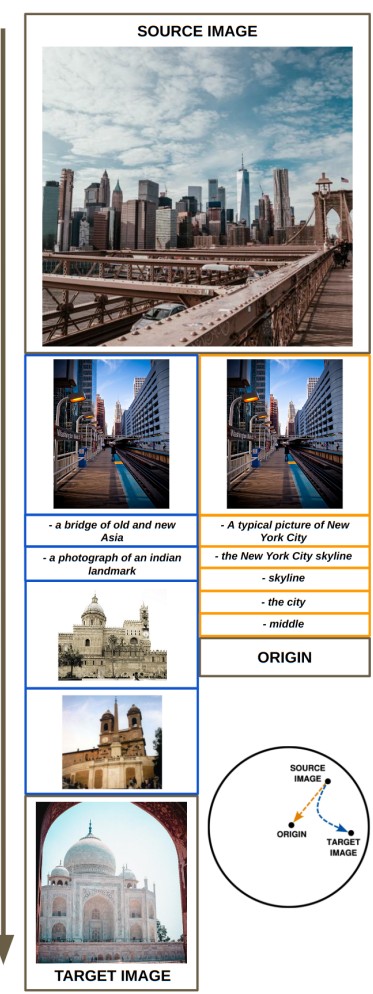

Figure 6: **Interpolation between points.** Multimodal retrieval results when moving from (*top*) an image to (*left*) another image or (*right*) the origin, as depicted in the (*bottom-right*) circle.

100 points being interpolated between two randomly selected images from `pexels.com` as well as to the origin. We observe that HyCoCLIP can fetch representations of both data modes in a very rational hierarchy. More interpolation examples are added in Appendix I where we also compare interpolation in MERU and CLIP representation space.

## 5 Related Work

**Vision-language models**   Currently expanding at a rapid pace, this topic has been in focus for multiple decades with initial works in image retrieval (Mori et al., 1999), semantic segmentation (Barnard et al., 2003), and object classification (Wang et al., 2009) leveraging natural language descriptors for computer vision. Later works (He & Peng, 2017) utilized more expressive representations from multi-modal neural network encoders. The advent of transformers (Vaswani et al., 2017) and vision transformers (Dosovitskiy et al., 2021) helped construct a highly semantic embedding space for texts and images, respectively. Recent works (Gan et al., 2022) have explored creating a shared embedding space by leveraging various pre-training strategies to integrate text and image information. Many approaches use contrastive learning as a core method, like CLIP (Radford et al., 2021) and ALIGN (Jia et al., 2021). More recently, MERU (Desai et al., 2023) combines entailment learning (Ganea et al., 2018a; Le et al., 2019) with the CLIP approach to learn embeddings in hyperbolic space capturing latent visual-semantic hierarchies. We extend this to include image patches and caption parts, enforcing an ordering that reflects the hierarchy shared by both modalities.

**Learning in hyperbolic space**   Hyperbolic space for representation learning has desirable properties for data with an inherent hierarchical or tree-like structure (Nickel & Kiela, 2017; Chamberlain et al., 2017). When generating embeddings in hyperbolic space from such data, its innate hierarchical structure can be retained with minimal distortion. As a result, hyperbolic deep learning has rapidly gained traction (Peng et al., 2022; Mettes et al., 2023). Recent works have developed methods for building neural networks that operate in hyperbolic space (Ganea et al., 2018b; Shimizu et al., 2021) and corresponding optimization algorithms (Bécigneul & Ganea, 2019; Bonnabel, 2013). This led to the use of hyperbolic models in many different modalities such as graphs (Liu et al., 2019; Franco et al., 2023; Flaborea et al., 2024), text (Dhingra et al., 2018; Tifrea et al., 2019), images (van Spengler et al., 2023; Atigh et al., 2022; Franco et al., 2024), videos (Long et al., 2020), time series (Flaborea et al., 2023), etc. Other recent work has focused on combining embedding spaces of different modalities (Liu et al., 2020; Desai et al., 2023). Mandica et al. (2024) also explore scaling hyperbolic embeddings in vision-language models to billions of parameters while maintaining stability and meaningful uncertainty estimates. Our work similarly learns multimodal representations in hyperbolic space to benefit from its inductive hierarchical bias.

**Hierarchies in vision and language**   Vendrov et al. (2016) use a visual-semantic hierarchy over words, sentences, and images to learn representations in a supervised fashion. They consider hypernym-hyponym relations in language to construct a hierarchy. This concept has been used in hypernymy detection tasks (Nguyen et al., 2017; Vulic & Mrksic, 2018). Hierarchies formed by constituency-based parse trees have been used to learn embeddings in hyperbolic space by Dhingra et al. (2018). In vision, several works sought to connect scenes to objects and parts of objects within the scene. Early works have used such information for pose estimation, image segmentation, and object and contour detection (Bourdev & Malik, 2009; Arbeláez et al., 2011). Recently, un-/self-supervised methods have been used for representation learning leveraging hierarchical segmentation of an image by Zhang & Maire (2020) and object-scene hypernymy by Xie et al. (2021); Ge et al. (2023). We combine hypernymy relations of vision and language.

## 6 Conclusion

The idea of this work is to use object compositions within a scene and its description, along with the visual-semantic ordering between image and text to learn hyperbolic representations that are semantically and hierarchically aligned. Our proposed HyCoCLIP improves over standard CLIP and its recent hyperbolic extension MERU in zero-shot classification. Moreover, our approach has increased scene understanding and better hierarchical structuring. Further, we qualitatively analyze the space by visualizing representations and through point-to-point interpolation which substantiates HyCoCLIP's ability to embed multi-modal hierarchies in a shared space. The method has certain limitations, with a key challenge being the need to generate bounding box information from image-caption pairs during training. This increases the volume of visual and textual data processed by HyCoCLIP, though it still preserves scalability during inference. Additionally, while our hierarchical training strategy improves interpretability by separating images and texts into distinct regions in the embedding space, it may not be optimal for tasks like large-scale retrieval.

ACKNOWLEDGEMENTS

We sincerely acknowledge the financial support given to Avik Pal by the Amsterdam ELLIS unit for the research and presentation of this paper at the conference. Max van Spengler acknowledges the University of Amsterdam Data Science Centre for financial support. We also extend our gratitude to SURF (`www.surf.nl`) for granting compute resources from the National Supercomputer Snellius. We are grateful to Panasonic for partially supporting this work. We acknowledge the financial support from the PNRR MUR project PE0000013-FAIR and from the Sapienza grant RG123188B3EF6A80 (CENTS). Thanks CINECA and the ISCRA initiative for high-performance computing resources and support.

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

## A    GENERATING BOX INFORMATION

We derive box information using an image grounding pipeline similar to Peng et al. (2023). Given an image-caption pair, noun entities are initially extracted from the caption into a list using spaCy (Honnibal et al., 2020). To minimize noise, we remove abstract nouns such as {*life, humor, love, ...*} from the list. We then predict the bounding boxes of the extracted entities within the image using the pre-trained grounding model GLIP (Li et al., 2022; Zhang et al., 2022). We exclude boxes with sizes lower than $32 \times 32$. We also threshold the predictions to at least 0.65 CLIP confidence score for the generated bounding box with the corresponding noun entity. Image-caption pairs for which no boxes could be generated or retained while filtering, are dropped. Further, referring expressions for noun chunks taken from the dependency tree of the caption using spaCy, are also included as text boxes. This increases the robustness of the stem towards linguistic complexities. A few samples from the GRIT dataset are visualized in Fig. 7.

## B    IMPLEMENTATION DETAILS

**Model architecture**    We use a similar setup as Desai et al. (2023), where the language encoder is the same one used by the original CLIP (Radford et al., 2021) consisting of a 12-layer Transformer architecture (Vaswani et al., 2017) with a width of 512 dimensions. The maximum input token size is set to 77 with a vocab size of 49408. For the vision encoder, we use the small and base Vision Transformer (Dosovitskiy et al., 2021; Chen et al., 2021; Touvron et al., 2021) backbone using a patch size of 16. The images are resized using border padding and random cropping (with scale $[0.5, 1.0]$) to $224 \times 224$, which results in an input sequence size of 196. A fixed set of 2-D sine-cosine position embeddings is included in the input sequence to instill a positional inductive bias.

**Initializing Lorentz model and Loss**    We train HyCoCLIP with a fixed curvature value of the Lorentz model on the grounded CC3M dataset for 40k steps. To find the optimal setting, we additionally

Table 7: **Training HyCo-CLIP on various settings of** $\kappa$ **on grounded CC3M dataset.** *dnc* denotes *did not converge*.

| $\kappa$ | ImageNet |
|---|---|
| Fixed param. | |
| 1.0 | 15.2 |
| 0.8 | *dnc* |
| 0.6 | 14.7 |
| 0.3 | 15.1 |
| 0.1 | 16.0 |
| Learnable param. | |
| **1.0** | **16.7** |

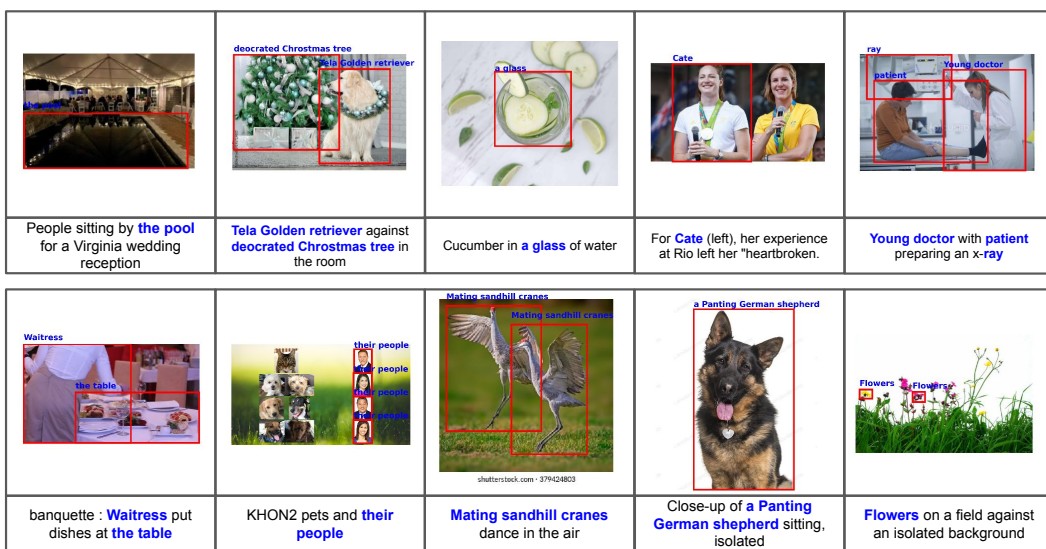

Figure 7: **Samples from the GRIT dataset.** The GRIT dataset (Peng et al., 2023) contains 20.5 million grounded vision-language pairs of which 10 random samples are visualized in this illustration. The grounded information includes noun terms or their referring expressions (highlighted in *blue*) and the corresponding bounding boxes (in *red* within the image).

train by keeping the curvature a learnable parameter with an initial value of $\kappa = 1.0$ and clamped within $[0.1, 10.0]$. As shown in Table 7, keeping the curvature a learnable parameter yields the best performance. Similar to Desai et al. (2023), we scale our batch of vectors before projecting it to the hyperboloid using learnable scalars $c_{img}$ and $c_{txt}$, respectively, in both image and text modes. These scalars are initialized with a value of $c_{img} = c_{txt} = 1/\sqrt{512}$. The adaptive softmax temperature of the contrastive loss is initialized with $\tau = 0.07$ and clipped at $0.01$. All of these scalar values are learned in the logarithmic space.

In the hCE loss (Equations 10,11), we set separate values of the $\eta$ parameter for inter-modality entailments $\eta_{inter} = 0.7$ and intra-modality entailments $\eta_{intra} = 1.2$ through a hyperparameter search while pre-training on the CC3M dataset for 75k steps and evaluating on ImageNet zero-shot image classification (cf. Table 8). Intuitively, this is because embeddings of images and text exist in different regions of the space, making it easier for text to entail the corresponding image as texts are nearer the origin and have a wider aperture $\omega$ (cf. Eq. 7). Hence, we make the loss stricter by reducing the aperture of text embeddings. Similarly, the intra-modal box representations are closer in their corresponding spaces. Accordingly, we increase the aperture of the box regions to relax the entailment loss. In the final hC loss, we set the weight for hCE loss $\gamma = 0.1$.

Table 8: **Hyperparameter search for $\eta_{inter}$** performed on grounded CC3M dataset.

| $\eta_{inter}$ | ImageNet |
|---|---|
| 1.0 | 12.5 |
| 0.9 | 12.6 |
| 0.8 | 13.1 |
| **0.7** | **13.4** |
| 0.6 | 13.3 |
| 0.5 | 12.8 |

**Optimizer and Hardware**   We train our models on 4 A100 GPUs for 500k steps using a batch size of 768 on an internal cluster. Similar to Desai et al. (2023), we use the AdamW optimizer (Loshchilov & Hutter, 2019) with hyperparameters $\beta_1 = 0.9, \beta_2 = 0.98$ and weight decay $0.2$ which is disabled for the learnable scalars. We use a cosine learning rate scheduler (Loshchilov & Hutter, 2017) with a maximum learning rate of $5 \times 10^{-4}$ and a linear rate for the initial 4k steps.

## C  METRICS FOR HIERARCHICAL CLASSIFICATION

This section provides more details on the metrics used for our hierarchical classification experiment. For a pair of predicted and true class $(\hat{y}, y)$, the Tree Induced Error (TIE) (Dekel et al., 2004) is the distance between $\hat{y}$ and $y$ in the graph (cf. Fig. 8a). This is defined as $\sum_{e \in E(\hat{y}, y)} w_e$, where $E(i, j)$ is the set of edges with weights $w_e$ along the path connecting nodes $i$ and $j$. For the WordNet graph, we set $w_e = 1$. Similarly, the Lowest Common Ancestor (LCA) error is the distance to the deepest common node in the graph which is shared between the ancestors of classes $\hat{y}$ and $y$.

For set-based measures, we define $\hat{Y}_{anc}$ and $Y_{anc}$ as the set of ancestor nodes of classes $\hat{y}$ and $y$ respectively (cf. Fig. 8b). Other relevant set-based hierarchical metrics such as, Jaccard Similarity $J$, and Hierarchical precision $(P_H)$ and recall $(R_H)$ (Kosmopoulos et al., 2015) are then given by

$$J = \frac{|\hat{Y}_{anc} \cap Y_{anc}|}{|\hat{Y}_{anc} \cup Y_{anc}|}, \quad P_H = \frac{|\hat{Y}_{anc} \cap Y_{anc}|}{|\hat{Y}_{anc}|}, \quad R_H = \frac{|\hat{Y}_{anc} \cap Y_{anc}|}{|Y_{anc}|}. \tag{13}$$

## D  REGIONCLIP ON OTHER TASKS

In addition to object detection (cf. Sec. 3.2, **Zero-shot object detection**), we evaluate RegionCLIP on the other downstream tasks described in Sec. 3.2. We compare RegionCLIP, which uses a ResNet-50 backbone ($\sim$25M parameters), to HyCoCLIP, which employs a ViT-S/16 backbone with $\sim$22M parameters. As shown in Table 9, our method significantly outperforms RegionCLIP across all these tasks. This is expected, as RegionCLIP is primarily optimized for object detection. By contrast, HyCoCLIP reaches the level of performance of RegionCLIP on object detection despite being designed to learn hierarchical representation spaces.

## E  SCENE UNDERSTANDING BENCHMARKS

In this section, we describe in detail the experiments of the compositional reasoning benchmarks used to evaluate our models in Sec. 3.2.

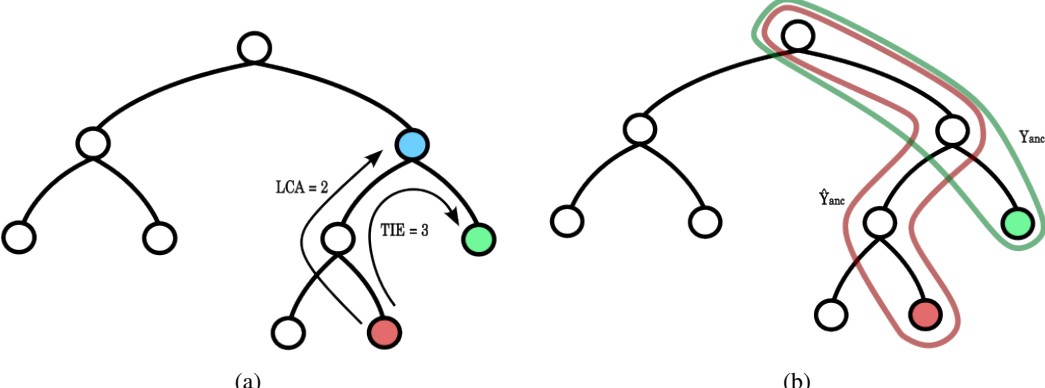

(a)                                                (b)

Figure 8: **Hierarchical classification metrics** when labels have a tree-like structure. Predicted label in *red* and true label in *green*. (a) Tree Induced Error (TIE) is the graph distance till the correct label while Least Common Ancestor (LCA) is the distance till the first common ancestor label as shown in *blue*. (b) shows ancestor label set of the predicted label ($\hat{Y}_{anc}$) and the true label ($Y_{anc}$).

Table 9: **RegionCLIP performance on downstream tasks.** The pre-training of RegionCLIP using boxes is primarily optimized for object detection. Thus displays poorer performance in comparison to HyCoCLIP which is designed to learn hierarchical representations from the boxes. * s.u. denotes scene understanding.

| | classification | | | | | retrieval | | hierarchical metrics | | | | | s.u.* | |
| Model | ImageNet | CIFAR-100 | SUN397 | Food-101 | Mean(16) | Text | Image | TIE ($\downarrow$) | LCA ($\downarrow$) | $J$ ($\uparrow$) | $P_H$ ($\uparrow$) | $R_H$ ($\uparrow$) | VL-CO | VG-A |
|---|---|---|---|---|---|---|---|---|---|---|---|---|---|---|
| RegionCLIP | 40.6 | 23.2 | 43.4 | 41.3 | 36.4 | 38.5 | 31.5 | 3.76 | 2.29 | 0.77 | 0.84 | 0.84 | 52.5 | 59.7 |
| HyCoCLIP | **41.7** | **53.6** | **52.5** | **50.2** | **41.1** | **69.5** | **55.2** | **3.55** | **2.17** | **0.79** | **0.86** | **0.85** | **59.8** | **68.4** |

### E.1 BENCHMARKS

**VL-Checklist-Object (VL-CO)**   This benchmark (Zhao et al., 2022) modifies the caption in several aspects. An object term in the caption is replaced with a random noun phrase. The model results are categorized for different sizes and locations of the object within the image to check for invariance which are summarized as follows,

- **O-Small**: The object covers a small area within the image. Following Zhao et al. (2022), the threshold of the object area is set to below $32 \times 32$.
- **O-Medium**: The object covers a moderate area within the image. The threshold of the object area is set between $32 \times 32$ and $96 \times 96$.
- **O-Large**: The object covers a large area within the image. Any object with an area greater than $96 \times 96$ fits this category.
- **O-Center**: The object center lies within the center region of the image. If $x$ is the half-length diagonal, and $y$ is the distance between the center of the object and the center of the image, the object is considered to lie in the center region if $\frac{y}{x} \leq \frac{1}{3}$.
- **O-Margin**: The object center lies at the margin of the image. This is when $\frac{y}{x} > \frac{2}{3}$.
- **O-Mid**: The object center lies in between the center and margin region which is when $\frac{1}{3} < \frac{y}{x} \leq \frac{2}{3}$.

**VG-Attribution (VG-A)**   This benchmark (Yüksekgönül et al., 2023) tests the capability of the model to correctly identify the attribute word associated with an object in a sentence in the context of an image. For instance, the model has to pick between "the **crouched cat** and the **open door**" and "the **open cat** and the **crouched door**".

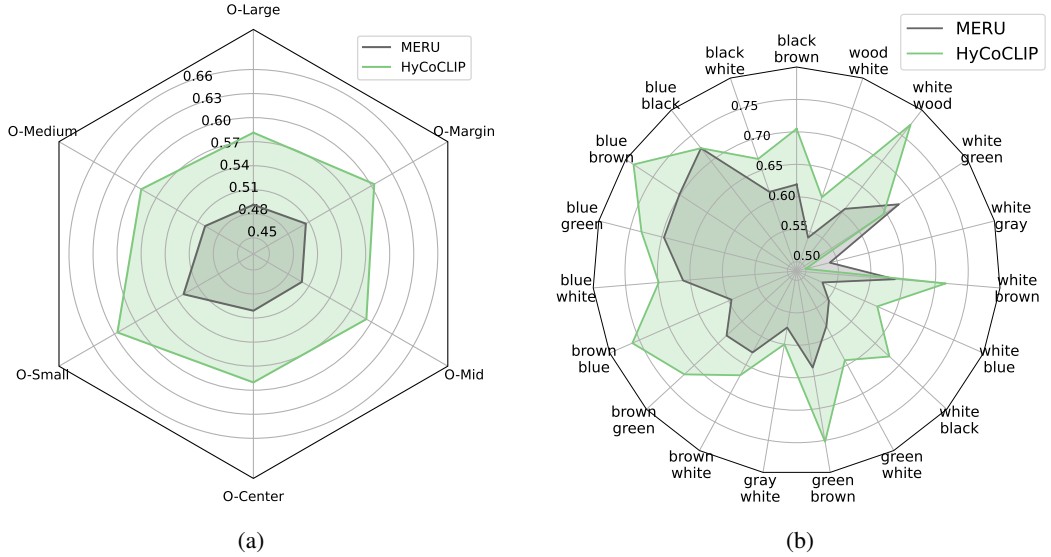

(a)  (b)

Figure 9: **Performance on scene understanding benchmarks** (a) model performance on VL-Checklist-Object (VL-CO) understanding experiments. HyCoCLIP performs best on object understanding tasks. (b) model performance on Visual Genome (VG) Attribution benchmark, accuracy values are plotted for the 19 most frequent attribute pairs in the experiment. HyCoCLIP gives the best results on most of these attribute pairs.

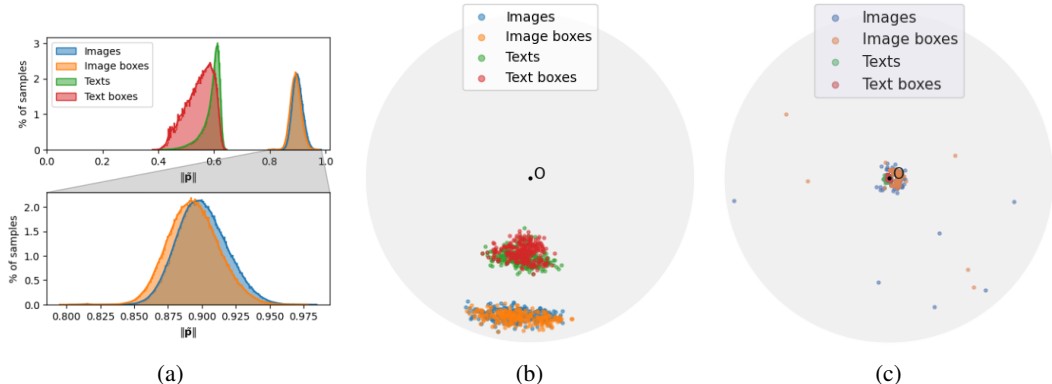

(a)  (b)  (c)

Figure 10: **Visualizing the learned hyperbolic space of MERU in lower dimensions** using samples from GRIT. (a) distribution of embedding distances from the origin, MERU embeds text data closer to the origin *wrt* the images but box samples don't show a smaller radius *wrt* full images/captions. On the right, (b) HoroPCA does not show local ordering and (c) CO-SNE visualizations of the latent space in $\mathbb{L}^2$ are quite distorted.

## E.2 Performance

We reported the mean performance for scene understanding benchmarks in Table 5. In addition to this, here we provide the results of HyCoCLIP compared against MERU on the individual categories of VL-CO and the top 19 most frequently occurring attribute pairs in the VG-A evaluation set. Fig. 9 highlights the significant improvements achieved by our method indicating better semantic scene understanding.

## F   VISUALIZING REPRESENTATION SPACE - MERU

Following Sec. 4, we additionally plot the learned hyperbolic space of MERU in lower dimensions in Fig. 10. We observe that the distributions of text embeddings and image embeddings are overlapped with corresponding box distributions from Fig. 10a. This is also apparent in the 2D plot when using HoroPCA in Fig. 10b. The embeddings for all modes seem to collapse to a small region with CO-SNE as seen in Fig. 10c.

## G   INTERPOLATION DETAILS

The logarithmic map is the inverse of the exponential map and, for $\mathbf{p}, \mathbf{q} \in \mathbb{L}^n$, is given by

$$\log_{\mathbf{p}}^{\kappa}(\mathbf{q}) = \frac{\cosh^{-1}(-\kappa \langle \mathbf{p}, \mathbf{q} \rangle_{\mathbb{L}})}{\sqrt{(\kappa \langle \mathbf{p}, \mathbf{q} \rangle)^2 - 1}}(\mathbf{q} + \kappa \langle \mathbf{p}, \mathbf{q} \rangle_{\mathbb{L}} \mathbf{p}). \tag{14}$$

Here, we will use it to interpolate between points in hyperbolic space (Desai et al., 2023).

For hyperbolic representations $(g_I(I_S), g_I(I_T))$ of source image $I_S$ and target image $I_T$, we take the logarithmic maps $\log_0(g_I(I_S))$ and $\log_0(g_I(I_T))$ and obtain a set of $N$ equally spaced representations on the line joining these vectors given by,

$$S_E^N = \left\{ p_i \in \mathcal{T}_p \mathbb{L}^n : p_i = (1 - t_i) \log_0(g_I(I_S)) + t_i \log_0(g_I(I_T)), \ t_i = \frac{i}{N}, \ i \in \{1, \dots, N\} \right\}. \tag{15}$$

These are then mapped back to the hyperboloid using exponential mapping given by,

$$S_H^N = \left\{ q_i \in \mathbb{L}^n : q_i = \exp_0(p_i), \ p_i \in S_E^N \right\}. \tag{16}$$

The closest representations are retrieved for all points in the set $S_H^N$ using Lorentzian distance as the similarity metric from a collection of representations of images and texts. Further, we drop any duplicate representations retrieved. For hyperbolic models, the origin ($\mathbf{0}$) of the space is taken as the *root* node, whereas for Euclidean CLIP, the *root* node is taken as the centroid of all the training samples.

## H   HIERARCHICAL IMAGE-TEXT MATCHING BENCHMARK

In this section, we evaluate our model and baselines on the hierarchical image-text matching benchmark using the HierarCaps (Alper & Averbuch-Elor, 2024) test set. Similar to the setup of Alper & Averbuch-Elor (2024) and Appendix G, we take 50 equally-spaced points between the root node and the nearest text embedding to the image embedding and calculate the precision (P) and recall (R) relative to the four hierarchically relevant ground-truth captions. In addition, we also report the order-aware metric $\tau_d$,

Table 10: **Hierarchical text-image matching** performed on HierarCaps benchmark.

| Model | P | R | $\tau_d$ |
|---|---|---|---|
| CLIP | 0.13 | 0.29 | 0.83 |
| MERU | 0.12 | 0.39 | 0.84 |
| HyCoCLIP | 0.12 | 0.46 | 0.88 |

which shows if the captions are ordered correctly in the embedding space. From the results in Table 10, we find HyCoCLIP outperforms CLIP and MERU, further demonstrating the enhanced hierarchical understanding achieved with our method. Further, from Fig. 11 we empirically find HyCoCLIP performs better hierarchical image-text matching on HierarCaps.

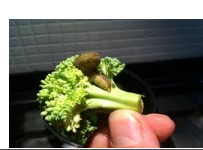

ROOT

↓

IMAGE

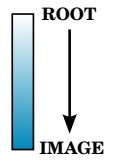

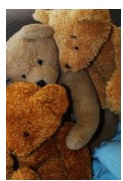

| HyCoCLIP | MERU | CLIP |
|---|---|---|
| remark | something | looking |
| out | image | legume |
| something | looking | opening |
| crop | photograph | variety |
| good | a picture of food . | crop |
| cooking | eukaryotic food | bud |
| savory | fresh food | a blinder that has some vegetables in it |
| vegetable | legume | lettuce |
| some vegetables | plant vegetable | A opened peeld banana half way eatened. |
| green veggie | some gross ass vegetables that look slimy AF | |
| edible vegetable | edible vegetable | |
| broccoli | herb | |
| a hand holding a vegetable | fried vegetables | |
| plate of broccoli | a blinder that has some vegetables in it | |
| a piece of broccoli | a hand holding a vegetable | |
| fresh broccoli with a salad | A raw piece of broccoli with something growing from it. | |
| finger holding a broccoli | | |
| The broccoli platter in the dish is mostly eaten up. | Two fingers holding a small piece of broccoli | |
| A worm sits on top of a piece of broccoli. | finger holding a broccoli | |

| HyCoCLIP | MERU | CLIP |
|---|---|---|
| remark | something | looking |
| out | picture | behind |
| something | looking | soft |
| miles | good | joes |
| little | people | stuffed |
| stuffed | baking food | teddy |
| some stuff | edible snack | stuffed bear |
| junk | sugary sweet | stuffed animal in a food |
| tiny room | affection | teddy bears |
| toys | two | a stuffed bear with others |
| teddy | fried fried | close up of stuffed animals . |
| stuffed bear | natural fiber | a couple of brown teddy bears |
| a little stuffed bear | stuffed | the teddy bears lying down on a quilt |
| stuffed bears | teddy | Three stuffed bears hugging and sitting on a blue pillow |
| a stuffed teddy bear sitting | a little stuffed bear | |
| a pair of stuffed teddy bears | stuffed teddy bear | |
| a stuffed bear with others | stuffed teddy bears | |
| Two stuffed teddy bears resting next to each other | stuffed bears | |
| a stuffed bear and a teddy bear | a pair of stuffed teddy bears | |
| Three teddy bears, each a different color, snuggling together. | a stuffed bear with others | |
| Three teddy bears laying in bed under the covers. | THIS IS A CLOSE UP PICTURE OF A STUFFED BEAR AND MONKEY | |
| Two teddy bears are laying side by side on a quilt. | a couple of brown teddy bears | |
| There are two stuffed bears and one of them is wearing a shirt. | teddy bears resting next to each other | |
| Two stuffed animals sit at a table with honey. | Three teddy bears, each a different color, snuggling together. | |

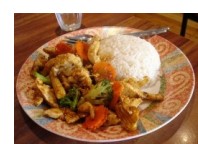

| HyCoCLIP | MERU | CLIP |
|---|---|---|
| remark | something | looking |
| out | image | living |
| something | living thing | legume |
| instrumental | guy | serving |
| legume | a picture of food . | curmudgeon |
| serving | savory food | food |
| good | The asian dish is filled with several different ingredients. | food served |
| cooking | a dish made from food on a plate | hot meal |
| dish | meal | a plate with cooked food |
| dinner | a plate of food . | person & dish with fajita on plate . |
| fried | the meal on a plate | a plate with cooked rice , vegetables and chicken |
| cooked food | a plate of food in a cooking . | |
| rice | a plate with cooked food | |
| hot meal | an image of a plate of food with meat and veggies | |
| chicken meal | person & dish with fajita on plate . | |
| the dish on a plate | A plate of vegetables, chicken, and white rice. | |
| the meal on a plate | A plate of chicken and vegetables sits next to a bowl of rice. | |
| a meal with some ingredients | | |
| a plate with cooked food and meat | | |
| a plate with cooked food | | |
| a plate with cooked rice , vegetables and chicken | | |
| A dinner plate that has white steamed rice with stir fry vegetables and chicken. | | |
| A plate of vegetables, chicken, and white rice. | | |
| A good lookign dish of food is arrange don a plate | | |
| A plate that has food that is sitting on the table. | | |

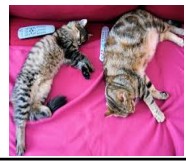

| HyCoCLIP | MERU | CLIP |
|---|---|---|
| remark | something | looking |
| bef | picture | soft |
| ohone | looking | pols |
| little | plural | lying |
| frizbee | domestic animal | furry |
| two | dilute | his fright |
| booie | cat | sleeping |
| lying | kitty | cats |
| furry | her cat | There are pets such as digs that are sleeping in the bef |
| kitty | kitten | pets sleeping in the bef |
| feline | cub | sleeping cats |
| cat laying down | feline | cats sleeping |
| kitty sitting | pets sleeping in the bef | cats sleeping with a remote |
| cat lying down | sleeping cat | the cats rest on a comforter |
| sleeping cat | There is a cat lying down on something. | Two cats rest on each other to take a nap. |
| sleeping cats | kitty sleeping | Two cats lay together on a blanket. |
| cats resting | sleeping cats | A couple of cats laying on top of a pink blanket. |
| cats sleeping | the cats are lying down | |
| cat resting on each other | cats sleeping with a remote | |
| Two cats laying together on a bed or floor | the cats rest on a comforter | |
| there are two cats that are laying with each other | Two cats lay together on a blanket. | |
| Two cats lay together on a blanket. | two cats lying near two remotes on a purple sheets | |
| A couple of cats laying on top of a pink blanket. | A couple of cats laying on top of a pink blanket. | |
| A young cat on a mat with a flip flop shoe. | | |
| two cats lying near two remotes on a purple sheets | | |

Figure 11: **Hierarchical image-text matching qualitative results on HierarCaps test set.** The color gradient reflects the concept's radius from the root of the space.

# I MORE INTERPOLATION EXAMPLES

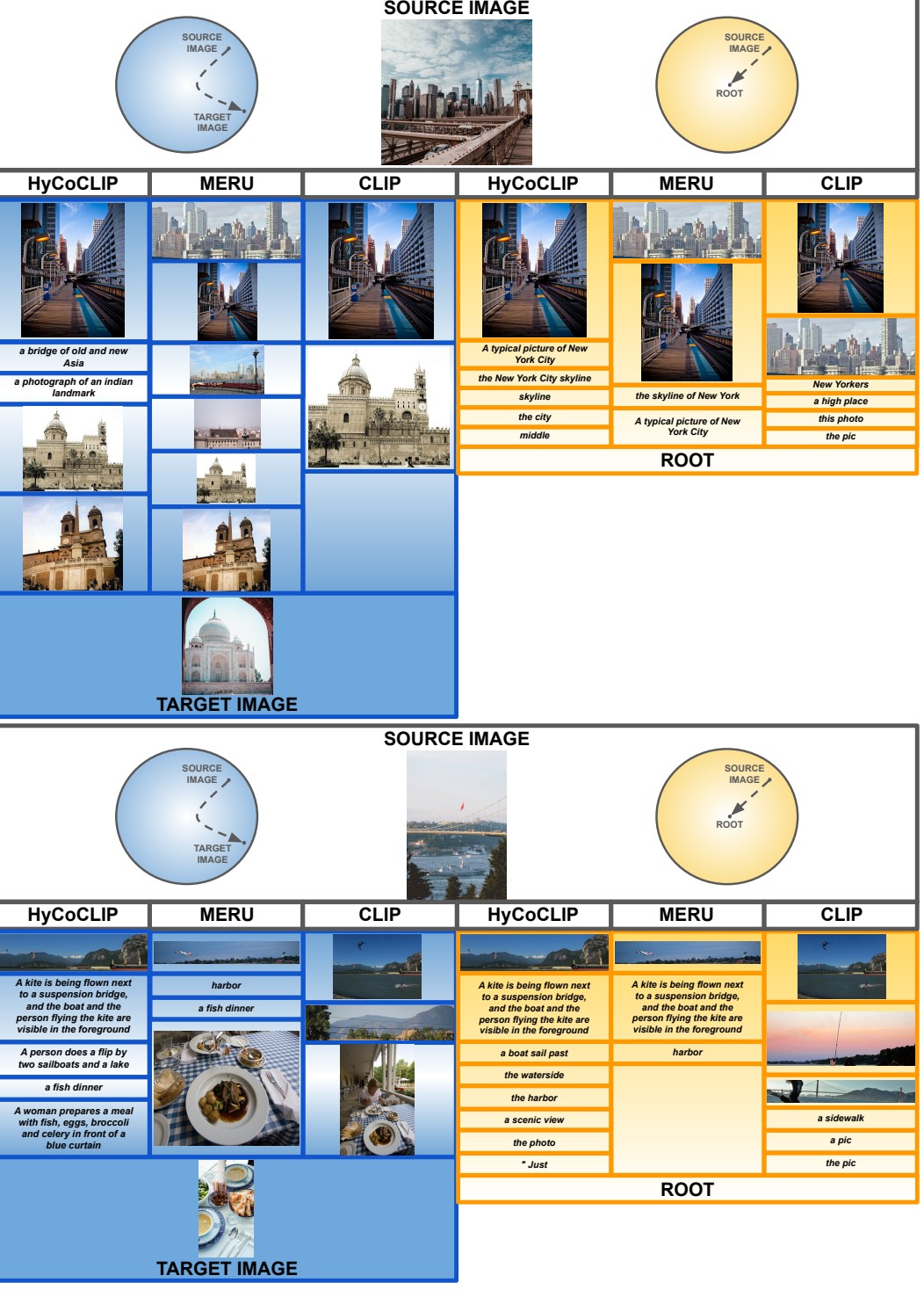

Figure 12: **Interpolation between points.** Multimodal retrieval results when moving from (*top*) an image to (*left*) another image or (*right*) the root. For HyCoCLIP and MERU, the root is the origin of the space, whereas it is the centroid of training sample representations for CLIP.

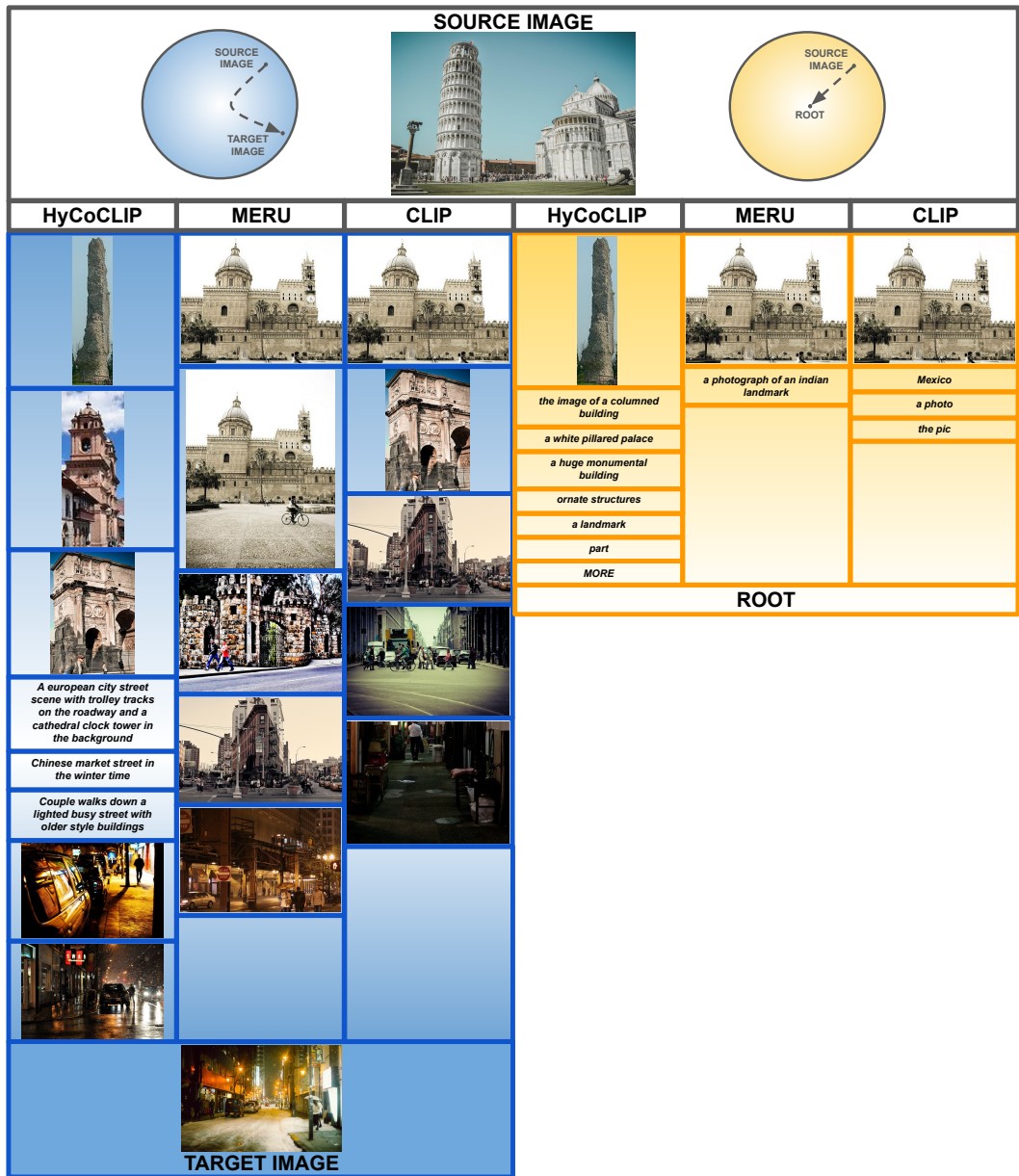

Figure 13: **Interpolation between points.** Multimodal retrieval results when moving from (*top*) an image to (*left*) another image or (*right*) the root. For HyCoCLIP and MERU, the root is the origin of the space, whereas it is the centroid of training sample representations for CLIP.

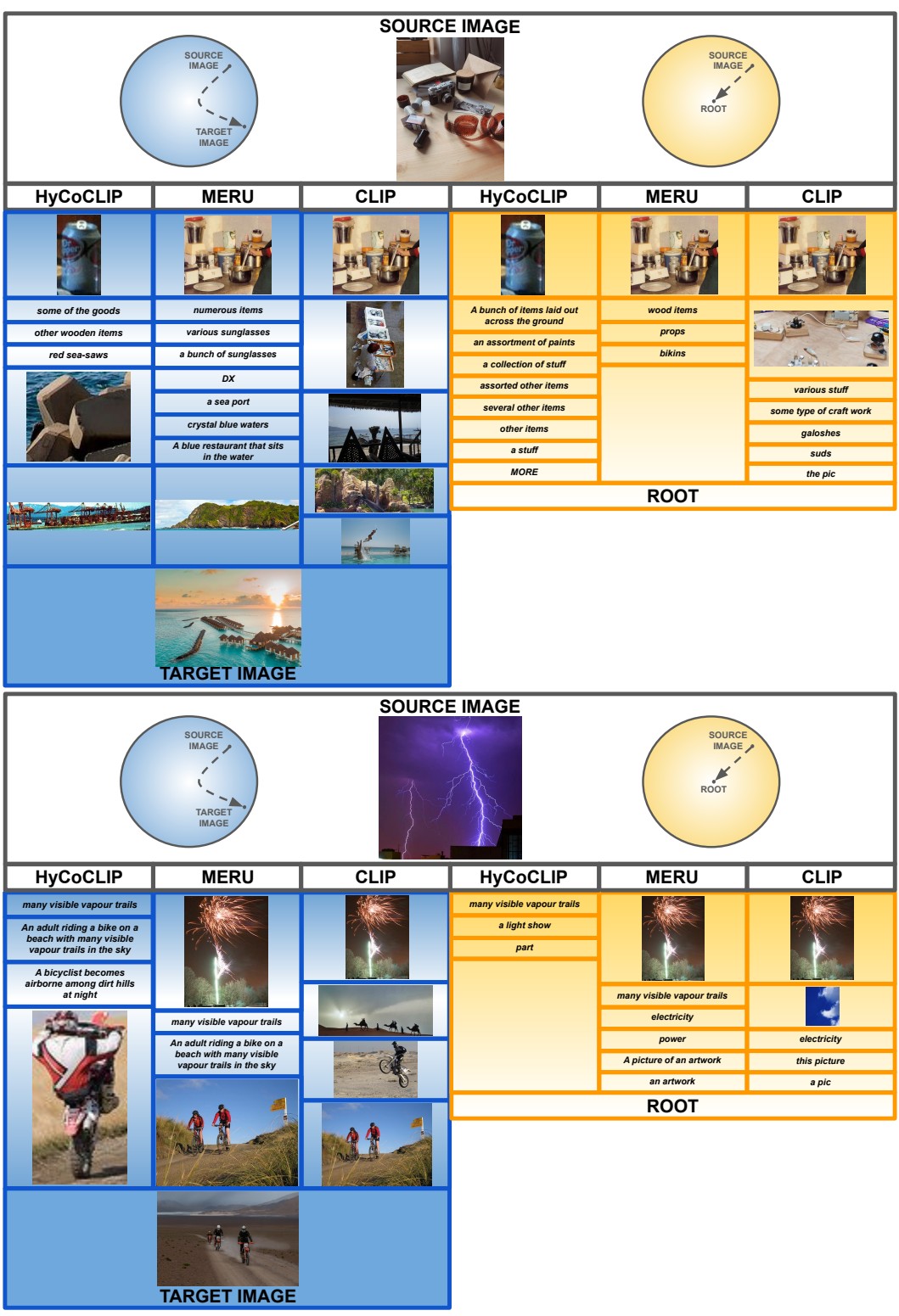

Figure 14: **Interpolation between points.** Multimodal retrieval results when moving from (*top*) an image to (*left*) another image or (*right*) the root. For HyCoCLIP and MERU, the root is the origin of the space, whereas it is the centroid of training sample representations for CLIP.

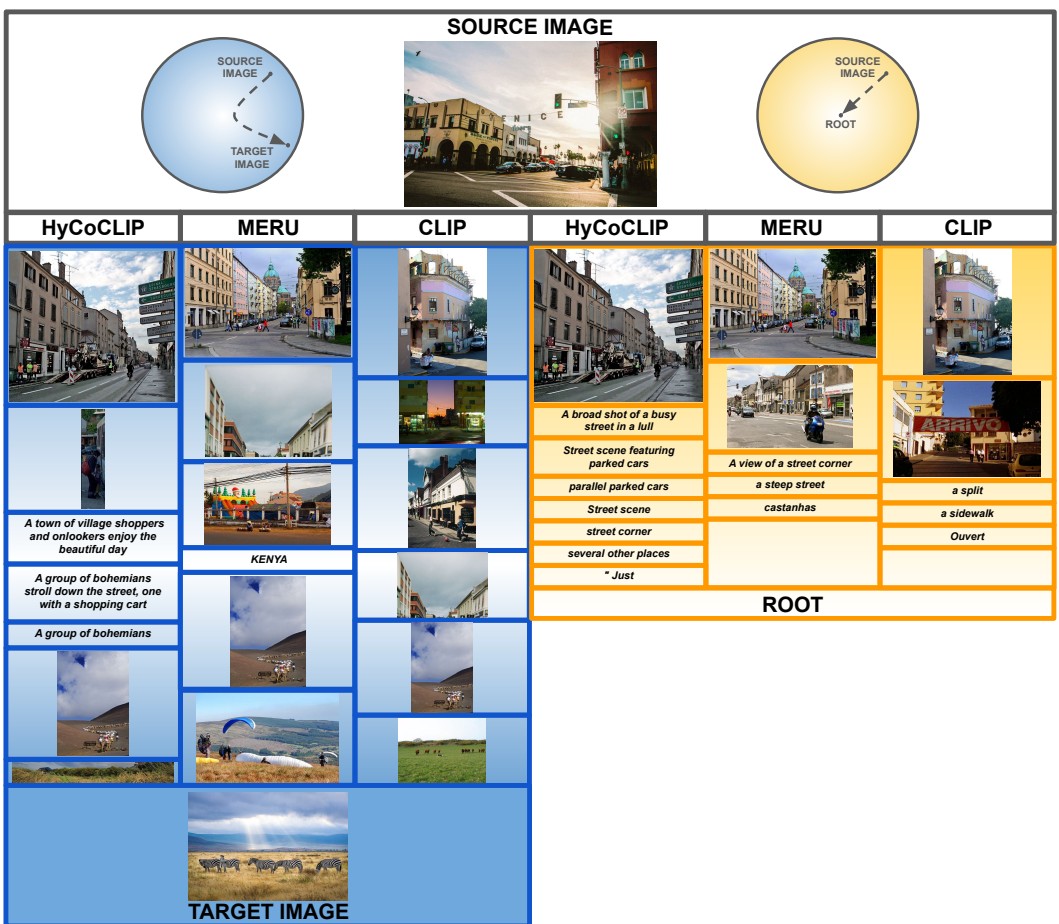

Figure 15: **Interpolation between points.** Multimodal retrieval results when moving from (*top*) an image to (*left*) another image or (*right*) the root. For HyCoCLIP and MERU, the root is the origin of the space, whereas it is the centroid of training sample representations for CLIP.

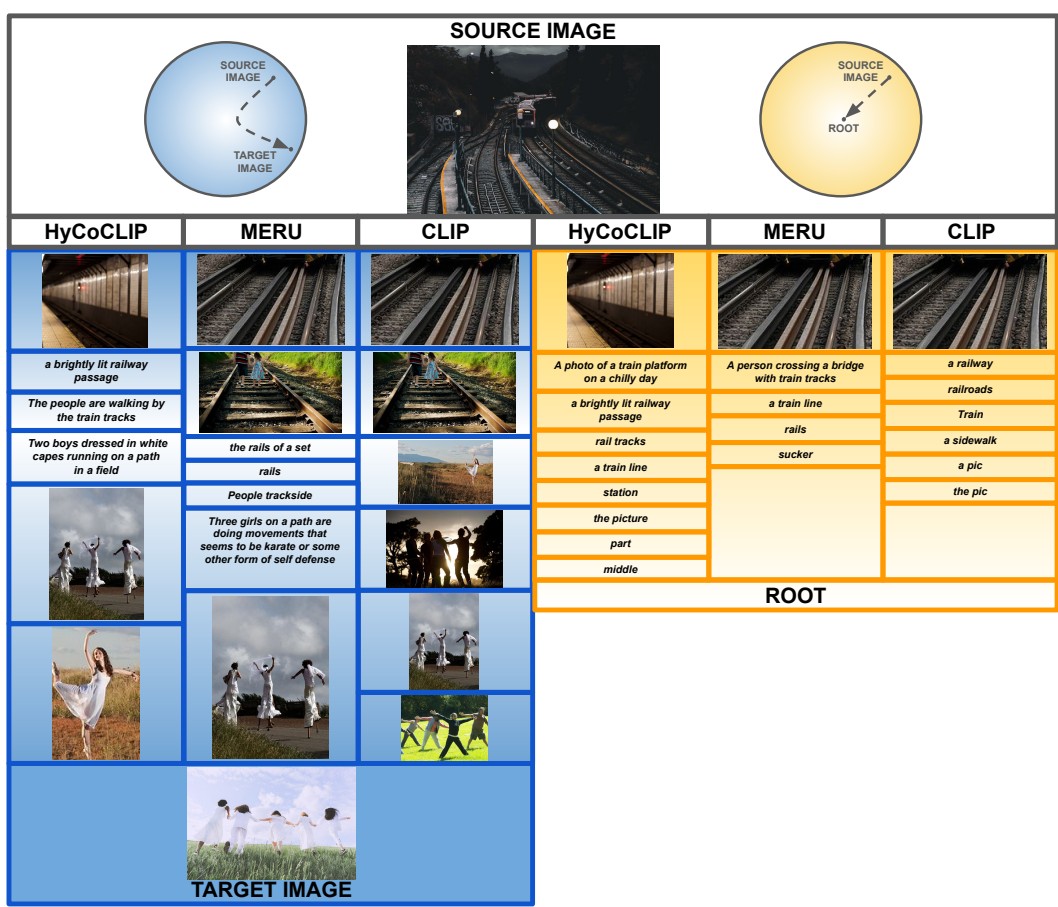

Figure 16: **Interpolation between points.** Multimodal retrieval results when moving from (*top*) an image to (*left*) another image or (*right*) the root. For HyCoCLIP and MERU, the root is the origin of the space, whereas it is the centroid of training sample representations for CLIP.

