# OpenReview forum: "Compositional Entailment Learning for Hyperbolic Vision-Language Models"
_ICLR.cc/2025/Conference — ICLR 2025 Oral_

### Official Review · Reviewer_5zgK · 2024-11-02

**Soundness:** 3
**Presentation:** 4
**Contribution:** 3
**Rating:** 8
**Confidence:** 3

**Summary:**

This paper proposes the novel Compositional Entailment Learning framework to train VLMs, by using as supervision the hierarchical relations between images, captions, and constituent nouns and their bounding boxes. Their results show that this outperforms standard CLIP and the hyperbolic CLIP variant MERU on both standard multimodal and hierarchical benchmarks. This is supported by qualitative results illustrating the learned hierarchical semantics of the learned space.

**Strengths:**

The central idea is clever and novel – utilizing the hierarchical nature of nouns mentioned in image captions as supervision for a hyperbolic model. The exposition is clear and concepts are well-illustrated. The quantitative experiments are extensive and overall convincing.

**Weaknesses:**

Qualitative results (Sec 4, Supp 8) are fairly limited. In particular, it is missing a qualitative comparison to existing models (CLIP, MERU) to illustrate whether HyCoCLIP’s embedding space represents hierarchies in a more qualitatively satisfying way.

While a comparison to CLIP trained from scratch is provided, recent work has found pretrained foundation VLMs to represent hierarchies in Euclidean space [1]. It would be useful to compare to such results to understand whether HyCoCLIP trained from scratch is competitive with such models.

[1] Alper and Averbuch-Elor. Emergent Visual-Semantic Hierarchies in Image-Text Representations. ECCV 2024

**Questions:**

Could the use of objects as supervision bias the model towards nouns and concrete concepts, possibly at the expense of attributes, dynamic actions (verbs), etc.?

Some details that are unclear from Supp. A: How were abstract nouns filtered? Are the nouns that can be grounded open-vocabulary (not limited to a fixed list)? How accurate is the GLIP-based grounding procedure?

---

> ### Author Response · Authors · 2024-11-23
>
> We thank the reviewer for their praise of the idea and their positive feedback regarding the presentation and the experiments.
>
> **Improved qualitative results.**
> We thank the reviewer for their suggestion and have now revised Appendix I by comparing HyCoCLIP traversals with both MERU and CLIP with plans to enrich the section with further illustrations. For CLIP, it should be noted that leveraging the Euclidean latent space, does not allow for a straightforward definition of the [ROOT] node. Therefore, for CLIP, we define this node as the centroid of the embeddings of the GRIT training samples, following the definition provided by MERU's authors.
>
>
> **Comparison to hierarchical understanding of pretrained foundation VLMs.**
> We thank the reviewer for the reference [1], which we will include in Sec. 5. We zero-shot evaluate our method on the introduced Hierarcaps test set of the reference paper on their proposed metrics. In the following table, we report the results along with baseline numbers from the reference (marked **).  We note that our models trained from scratch on a significantly lower volume of data perform comparably to the OpenCLIP and ALIGN baselines. Additionally, HyCoCLIP outperforms CLIP and MERU indicating better hierarchical understanding.
>
>
> | Model           | Precision | Recall | $\tau_{corr}$ |
> |-----------------|-----------|--------|---------------|
> | OpenCLIP**      | 0.16      | 0.33   | 0.87          |
> | ALIGN**         | 0.16      | 0.36   | 0.89          |
> |-----            |-----      |-----   |-----          |
> | CLIP-ViT-B      | 0.13      | 0.29   | 0.83          |
> | MERU-ViT-B      | 0.12      | 0.39   | 0.84          |
> | HyCoCLIP-ViT-B  | 0.12      | 0.46   | 0.88          |
>
> We have added further details on this experiment in our revised manuscript Appendix H.
>
>
> **Does the use of objects induce a bias towards nouns and concrete concepts?**
> Our scene understanding experiment, described in detail in Sec. 3.2 and Appendix E, includes the VL Checklist Object and VG Attribute benchmarks, which feature words and expressions distinct from standard noun concepts. We performed this zero-shot evaluation test to assess our model's ability to comprehend attributes like color (e.g., white, brown, orange), verbs (e.g., crouched, sitting, burnt), materials (e.g., wood, metal, mesh), adjectives (e.g., empty, young), as well as object size and spatial positioning within the image. The outcomes show that our proposed method improves on the MERU's technique [2] also in this challenging task.
>
>
> **Filtering nouns.**
> The abstract nouns are part of a predefined list that is excluded when extracting noun chunks using spaCy's English language model. The extracted nouns are largely open-vocabulary, as spaCy’s part-of-speech vocabulary is quite extensive. For grounding, the bounding boxes predicted by the GLIP model are filtered when their confidence score (measured by the dot product with the corresponding noun chunk) is below 0.65, ensuring that only high-quality boxes are considered. We have now included Fig. 7 in Appendix A which provides qualitative examples.
>
>
> [1] Alper and Averbuch-Elor. "Emergent Visual-Semantic Hierarchies in Image-Text Representations", ECCV 2024.

---

> > ### Comment · Reviewer_5zgK · 2024-11-24
> >
> > Thank you for your detailed response. This addresses my concerns and questions well and I believe the added details strengthen the paper. I will maintain my accept rating.

---

### Official Review · Reviewer_UEf2 · 2024-11-04

**Soundness:** 3
**Presentation:** 3
**Contribution:** 3
**Rating:** 8
**Confidence:** 3

**Summary:**

The authors proposed to incorporate hierarchical pretraining for hyperbolic vision language models and the resulting model Hyperbolic Compositional CLIP (HyCoCLIP). The core idea is to construct object regions (image boxes) and corresponding text phrases (text boxes) to build a multi-layered, compositional hierarchy within the shared hyperbolic embedding space. The HyCoCLIP shows competitive performance in zero-shot classification and retrievals. The author also conducted experiments to show how HyCOCLIP can outperform CLIP and the hyperbolic contrastive model MERU in zero-shot hierarchical classification and scene understanding tasks.

**Strengths:**

I think this paper is very well written and I find it easy to follow. Overall the idea behind HyCoCLIp is well motivated and I believe the authors have conducted sufficient experiments to empirically demonstrate the proposed method and model’s efficacy. The empirical performance of HyCoCLIP is very strong and to the best of my knowledge, the proposed HyCoCLIP achieved the state-of-results on many of the reported zero-shot tasks from a contrastive-pretrained model.

**Weaknesses:**

One major concern is the incremental nature of this work. Hyperbolic embeddings for representing hierarchical relationships have been explored in previous models, and this paper primarily builds upon these established ideas. However, the specific contributions of HyCoCLIP, particularly in enhancing hierarchical and scene understanding tasks, offer sufficient merit to make this work valuable to the broader community.

**Questions:**

In Table 1/2, the authors bold the best performance overall across different model backbones. Wouldn’t it be more informative and fair to bold the best performance within each backbone group (e.g., ViT-S/16, ViT-B/16) to allow for a clearer comparison of HyCoCLIP’s performance relative to baselines on similar architectures?

Regarding the choice of batch size, the authors used a batch size of 768 due to memory limitations. Did the authors consider implementing techniques like gradient accumulation to effectively simulate a larger batch size? This could provide further insights into how batch size impacts model performance, especially since batch size has been shown to affect contrastive learning tasks significantly.

---

> ### Author Response · Authors · 2024-11-23
>
> We are grateful to the reviewer for acknowledging the contributions made by our method towards enhancing hierarchical understanding in VLMs. While existing works mainly evaluate the hierarchical nature of representation space, we go further toward enforcing it through our novel loss functions while demonstrating improved performance on several downstream tasks. In the following, we address the questions raised by the reviewer.
>
>
> **Highlighting best performances.**
> We thank the reviewer for this suggestion. Following both their and reviewer SpcR's recommendation, we revised Table 1 to include the results of HycCoCLIP on the RedCaps dataset and underline the best results among competitors sharing the same backbone.
>
>
> **Gradient accumulation for simulating larger batch size.**
> We agree with the reviewer that gradient accumulation could effectively simulate larger batch sizes. However, in the final paragraph of Sec. 3.3, we address the impact of the batch size on the performance. Our findings indicate no noticeable benefit in using very large values for the batch size (cf. Table 6 in the main manuscript). This saturation with batch size in contrastive loss has also been discussed by [1] with the additional entailment loss also playing a factor. Thus, we did not focus on scaling to greater batch sizes.
>
>
> [1]: Zhai *et al.*, "Sigmoid loss for language image pre-training", ICCV 2023.

---

### Official Review · Reviewer_cUjb · 2024-11-07

**Soundness:** 4
**Presentation:** 3
**Contribution:** 3
**Rating:** 8
**Confidence:** 4

**Summary:**

This paper introduces a novel approach named HyCoCLIP to vision-language modeling that leverages the hierarchical nature of hyperbolic space to better align visual and textual data. It organizes image and text data as hierarchical compositions, where objects within an image and their corresponding text descriptions are represented at different levels of abstraction in hyperbolic space.The experiments demonstrate that HyCoCLIP achieves significant performance improvements across multiple tasks.

**Strengths:**

1. This paper is well-organized. The motivation is easy to follow, and the method is easy-to-understand.
2. The proposed HyCoCLIP is novel and effective. It organizes data at multiple abstraction levels, providing an inspiring approach to multi-modal learning.
3. The authors performs exhaustive experiments to show that the effectiveness of HyCoCLIP. It outperforms baselines on general and fine-grained image classification tasks.

**Weaknesses:**

1. While the paper compare with CLIP and MERU, it should also compare some recently proposed VLMs.
2. The paper should explore how sensitive the model is to the choice of hyperbolic space parameters.

**Questions:**

Could you please provide more details on the choice of hyperbolic space parameters?

---

> ### Author Response · Authors · 2024-11-23
>
> We thank the reviewer for their positive feedback regarding the organization of the paper, the method, and the experiments.
>
>
> **Comparison with recently proposed VLMs.**
> We compare our method with the recent SOTA hyperbolic VLM, i.e. MERU in addition to Euclidean CLIP. While there are indeed other recent Euclidean VLMs that excel at various tasks, these models are typically trained at a much larger scale, often requiring vast amounts of data and computational resources. As a result, a direct comparison between these models and our approach, which operates with a limited training setup and a relatively limited amount of data, would be impossible.
>
>
> **Details and sensitivity of hyperbolic space parameters.**
> The key geometric parameters in hyperbolic space are the curvature and dimensionality. We allow the curvature ($\kappa$) to be a learnable parameter (initialized at $\kappa=1$), consistent with prior work [1], and clamped the parameter at 0.1 value. In response to the reviewer’s suggestion, we have included Table 7 in Appendix B to demonstrate HyCoCLIP's sensitivity to this parameter. We find that enabling $\kappa$ to be learned while training empirically yields the best results.
>
> As for the dimensionality, we use the Lorentz model to represent the hyperbolic space $\mathbb{L}^n$ with $n=512$. In addition to these parameters, we experimented to determine the optimal aperture for entailment cones to maximize performance. The results of this analysis are detailed in Appendix B. While some of these details were already provided in Appendix B, we have expanded and clarified this section to include these further insights.
>
> [1]: Desai *et al.*, "Hyperbolic Image-Text Representations", ICML 2023.

---

### Official Review · Reviewer_SpcR · 2024-11-07

**Soundness:** 3
**Presentation:** 4
**Contribution:** 3
**Rating:** 8
**Confidence:** 3

**Summary:**

This work proposes a novel learning method for training vision-language models. Specifically, the method involves pretraining such models with 2 losses --- hierarchical compositional contrastive and entailment losses. The hierarchical concepts correspond to image boxes and the corresponding text boxes. The experiments are conducted on large scale dataset (GRIT) consisting of 20.5M image-text pairs. In Appendix A, the authors describe an automatic procedure to obtain the text boxes (noun entities in this case) and their corresponding bounding boxes in the images. The paper details empirical results on a variety of tasks including zero-shot image classification, retrieval, object detection and scene understanding.

**Strengths:**

* The proposed method is simple and elegant and can be easily applied to large scale pretraining of vision-language models. The procedure to automatically generate paired image and text boxes is also relatively straightforward.
* The empirical results show improvement across several tasks which demonstrates the improved representation learning - classification, retrieval, detection and understanding.
* Table 1 results show that CLIP trained on additional image-text boxes doesn't improve the performance. However, training on the same data but with the proposed hierarchical compositional learning losses shows significant improvement in performance. This further demonstrates the effectiveness of the proposed technique.

**Weaknesses:**

When training CLIP on additional image-text boxes shows no improvement (Table 1), it could be because there is limited new information in such examples (as original image-text pairs are already present in the training data). For a better understanding of this, an experiment such as this might help: split the GRIT dataset into 2 random subsets of 10M each. Then compare the results on the following settings:

[1] CLIP trained on 10M image-text pairs

[2] CLIP trained on 10M image-text pairs + additional image-text boxes

[3] HyCoCLIP trained on 10M image-text pairs + additional image-text boxes

[4] CLIP trained on 20M image-text pairs

The paper presents the comparison of [1] vs [2] vs [3] (but on all 20M image-text pairs) in Table 1 but comparing [3] vs [4] will help answer the above question. It is worth noting that even if the comparison shows similar results, [3] might still be slightly favored since it can be applied on top of any existing large dataset.

**Questions:**

Can the authors share the results of HyCoCLIP on RedCaps dataset?

---

> ### Author Response · Authors · 2024-11-23
>
> We appreciate the reviewer’s positive feedback on our method and the results achieved. Below, we address the questions raised.
>
>
> **Training CLIP on additional image-text boxes shows no improvement.**
> We agree with the reviewer that image-text boxes do not offer additional information when naively added as extra samples since they are directly extracted from the full image-text pairs. This is precisely where our approach excels, by leveraging the hierarchical alignment of additional boxes within the hyperbolic space. Additionally, we conducted the suggested experiment of training on a half-split of the training set on the ViT-S backbone. We present the results in the following tables and also include results of HyCoCLIP and CLIP on the entire GRIT as reported in Table 1 of our manuscript.
>
> Evaluation of the models on zero-shot image classification,
>
>
> | Model     | Pre-training | w/ boxes | samples (M) | ImageNet | Caltech-101 | Food-101 | Pets | RESISC45 | Mean(16) |
> |-----------|--------------|----------|-------------|----------|-------------|----------|------|----------|----------|
> | CLIP      | GRIT (Half)  | N        | 10          | 31.5     | 65.4        | 42.3     | 36.7 | 37.1     | 33.4     |
> | CLIP      | GRIT (Half)  | Y        | 10 [18.7]   | 29.9     | 70.0        | 38.3     | 36.5 | 31.9     | 35.2     |
> | HyCoCLIP  | GRIT (Half)  | Y        | 10 [18.7]   | 35.3     | 71.1        | 46.9     | 44.0 | 37.2     | 36.6     |
> |-----      |-----         |-----     |-----        |-----     |-----        |-----     |----- |-----     |-----     |
> | CLIP      | GRIT         | N        | 20.5        | 36.7     | 73.6        | 44.7     | 44.6 | 40.1     | 37.1     |
> | CLIP      | GRIT         | Y        | 20.5 [35.9] | 36.2     | 74.1        | 43.2     | 45.9 | 35.5     | 38.2     |
> | HyCoCLIP  | GRIT         | Y        | 20.5 [35.9] | 41.7     | 75.7        | 50.2     | 52.0 | 45.7     | 41.1     |
>
> $~$
>
> Evaluation of the models on Flickr image/text retrieval,
>
>
> | Model     | Pre-training | w/ boxes | samples (M) | Text | Image |
> |-----------|--------------|----------|-------------|------|-------|
> | CLIP      | GRIT (Half)  | N        | 10          | 86.8 | 76.8  |
> | CLIP      | GRIT (Half)  | Y        | 10 [18.7]   | 79.8 | 68.3  |
> | HyCoCLIP  | GRIT (Half)  | Y        | 10 [18.7]   | 87.4 | 77.7  |
> |-----      |-----         |-----     |-----        |----- |-----  |
> | CLIP      | GRIT         | N        | 20.5        | 90.2 | 81.1  |
> | CLIP      | GRIT         | Y        | 20.5 [35.9] | 84.2 | 73.1  |
> | HyCoCLIP  | GRIT         | Y        | 20.5 [35.9] | 89.1 | 81.5  |
>
> $~$
>
> We find similar trends to those presented originally in Table 1 of the manuscript. HyCoCLIP benefits the most from such localized boxes. Additionally, from the first table, we also find that HyCoCLIP scales best with the additional (nearly double) data.
>
>
> *(continued)*

---

> > ### Author Response · Authors · 2024-11-23
> >
> > **Results of HyCoCLIP on RedCaps.**
> > The revised manuscript now reports the figures for HyCoCLIP when trained with RedCaps in Table 1. For convenience, we also provide the results here. In the previous version, we did not report these numbers as the comparison was considered unfair since HyCoCLIP only has access to 5.8 M samples from the original dataset (to be compared with 11.4 M samples for MERU [1] and CLIP). However, the results are comparable (even better on a few) which further demonstrates that our technique is also effective in data-scarcity regimes due to the improved handling of the inter- and intra-modality hierarchies. We thank the reviewer for their suggestion and updated our proposed work.
> >
> > | Dataset     | Model        | samples (M) | ImageNet | CIFAR-10 | CIFAR-100 | SUN397 | Caltech-101 | STL-10 | Food-101 | CUB  | Cars | Aircraft | Pets | Flowers | DTD | EuroSAT | RESISC45 | Country211 |
> > |-------------|--------------|-------------|----------|----------|-----------|--------|---------|-----------|-------------|------|------|----------|------|---------|-----|---------|----------|------------|
> > |             | CLIP†        | 11.4        | 32.5     | 66.7     | 35.8      | 26.7   | 60.8    | 89.8      | 72.5        | 29.8 | 11.1 | 1.3      | 72.5 | 44.9    | 16.4| 30.1    | 27.7     | 5.0        |
> > |             | CLIP         | 11.4 [6.3]  | 30.2     | 76.5     | 42.4      | 25.8   | 62.3    | 89.5      | 69.6        | 25.7 | 8.5 | 2.2      | 65.3 | 38.6    | 13.6| 36.6    | 28.5     | 4.6        |
> > | **RedCaps** | MERU†        | 11.4        | 31.4     | 65.9     | 35.2      | 26.8   | 58.1    | 89.3      | 71.4        | 29.0 | 8.3  | 1.6      | 71.0  | 40.9    | 17.0| 29.9    | 29.3     | 4.7        |
> > |             | MERU         | 11.4 [6.3]  | 29.9     | 76.4     | 39.9      | 26.6   | 62.3    | 89.5      | 68.4        | 25.4 | 8.9  | 1.2      | 67.2  | 37.6    | 13.0| 30.5    | 27.6     | 4.3        |
> > |             | **HyCoCLIP** | 5.8 [6.3]   | **31.9** | **77.4** | **37.7**  | **27.6** | **64.5** | **90.9** | **71.1**    | **28.8** | **9.7** | **1.1**    | **70.5**  | **41.4**| **13.4** | **22.7**| **30.7**     | **4.4**        |
> >
> >
> >
> >
> >
> > [1]: Desai *et al.*, "Hyperbolic Image-Text Representations", ICML 2023.

---

> > > ### Comment · Reviewer_SpcR · 2024-12-02
> > >
> > > Thank you for running the additional suggested ablations. As mentioned in my review, this ablation presents a better understanding of the dynamics: the results show that training on all the image-text pairs without any additional loss (Row 4 in the table) is generally better than training on half the number of image-text pairs with additional hierarchically aligned image-text pairs along with proposed losses. In some sense, this shows an upper bound of the proposed method.
> > >
> > > As mentioned in the original review, this ablation is very helpful for further understanding but does not dilute the existing contributions and results. The efforts spent by authors on running additional experiments is appreciated. I will stay with my original rating of accept.

---

### Author Response · Authors · 2024-11-23

We thank the reviewers for their valuable feedback and for positively rating our work. We appreciated the shared comments about the clarity of the writing and the novelty of our proposed approach. In the revised manuscript, we implemented suggestions and added several details of HyCoCLIP that were recommended by the reviewers, which we think contribute to strengthening our work. These additions can be found in the revised manuscript, highlighted in red.

We have included several comprehensive qualitative results in Appendix I. Additionally, including illustrations from the GRIT dataset eases the understanding of the grounding procedure of Appendix A while also providing qualitative insights. We have also revised Table 1 following the reviewers' suggestions and report the sensitivity of HyCoCLIP to the curvature of the hyperbolic space in Table 7 of Appendix B. Additionally, the results of the zero-shot evaluation of HyCoCLIP on the HierarCaps dataset are in Table 10 of Appendix H.

In the following, we address the questions and weaknesses raised by the reviewers to the best of our understanding and remain open to additional feedback.

---

### Meta-Review · Area_Chair_4gPK · 2024-12-22

**Metareview:**

This paper studies the hierarchical visual-text representation. Concretely, the author proposed to leverage the hierarchical relation within the image (whole image and objects) and the text (whole sentence and nouns) to construct a hierachical embedding space, where the more general terms (objects / nouns) are pushing towards the origin and the more specific terms (sentences and whole images) are pushing towards the boundary. To construct this hierarchical embedding space, the author proposed object (noun) and sentence (image) contrastive loss and also entailment loss.

Strength:
1. The paper is easy to follow and well-written
2. The proposed approach is interesting and novel.
3. The proposed approach achieves good performance on multiple benchmarks.

Weakness:
1. Considering the related works on hierarchical and hyperbolic space, the proposed approach might be slightly incremental.

Given all the support from the reviewers (all 8), I recommend accept.

**Additional Comments On Reviewer Discussion:**

During the rebuttal, the reviewers agree this is a good paper and should be accepted.
The reviewers point out weakness:
1. More ablation are needed for understanding why additional image-text boxes is not helpful for CLIP but helpful for the proposed approach.
2. Sensitiveness of the choice of the hyperbolic embedding space
3. More qualitative and quantitative comparison with the recent VLM approaches.

All the reviewers agree that their concerns are well addressed. Scores are maintained as all 8.

---

### Decision · Program_Chairs · 2025-01-22

Accept (Oral)